# Development of an AI model for DILI-level prediction using liver organoid brightfield images
Shiyi Tan [1,2,5], Yan Ding [2,5], Wei Wang[1,2], Jianhua Rao[3], Feng Cheng[3], Qiuyin Zhang[2], Tingting Xu[2], Tianmu Hu[2], Qinyi Hu[2,4], Ziliang Ye[2,4], Xiaopeng Yan[2], Xiaowei Wang[2], Mingyue Li[4], Peng Xie [4], Zaozao Chen[4], Geyu Liang[1], Yuepu Pu[1], Juan Zhang [1,2] ✉ & Zhongze Gu [2,4] ✉

AI image processing techniques hold promise for clinical applications by enabling analysis of complex status information from cells. Importantly, real-time brightfield imaging has advantages of informativeness, non-destructive nature, and low cost over fluorescence imaging. Currently, human liver organoids (HLOs) offer an alternative to animal models due to their excellent physiological recapitulation including basic functions and drug metabolism. Here we show a drug-induced liver injury (DILI) level prediction model using HLO brightfield images (DILITracer) considering that DILI is the major causes of drug withdrawals. Specifically, we utilize BEiT-V2 model, pretrained on 700,000 cell images, to enhance 3D feature extraction. A total of 30 compounds from FDA DILIrank are selected (classified into Most-, Less-, and No-DILI) to activate HLOs and corresponding brightfield images are collected at different time series and z-axis. Our computer vision model based on image-spatial-temporal coding layer excavates fully spatiotemporal information of continuously captured images, links HLO morphology with DILI severity, and final output DILI level of compounds. DILITracer achieves an overall accuracy of 82.34%. To our knowledge, this is the first model to output ternary classification of hepatotoxicity. Overall, DILITracer, using clinical data as an endpoint categorization label, offers a rapid and effective approach for screening hepatotoxic compounds.

The computer vision (CV) model has shown significant potential in clinical applications by enabling detailed analysis of complex visual information from cell images[1]. Recently, vision transformer (ViT) has made breakthrough progress in the field of CV, signaling a transition from the Convolutional Neural Network to the Transformer backbone[2,3]. ViT utilizes a self-attention mechanism to capture long-distance relationships in an image, enabling it to understand global dependencies in the data[4]. This ability to capture the overall structure of biomedical images is why we selected ViT for this work. CV models using two-dimensional (2D) biomedical images have delivered impressive predictive capabilities in tasks such as detecting cell death[5], segmenting cell nuclei[6], and localizing subcellular protein[7]—achievements that are difficult with manual analysis. However, the emergence of physiologically relevant three-dimensional (3D) models like spheroids[8,9] and organoids[10], underscores the urgent need for the development of advanced 3D imaging techniques and novel cell morphology analysis algorithms in CV. Currently, while drug screening based on phenotypes or statuses from cell images has gradually been applied, there has been less focus on identifying drug-induced liver injury (DILI). This may be due to the substantial metabolic differences between humans and animals[11], making it difficult to reflect the actual effects of compounds. For example, traditional preclinical safety trials of 150 drugs reported predictive accuracies of only 63% and 43% in non-rodent and rodent animals, respectively, with the lowest accuracy observed in the hepatobiliary system[12]. Preclinical identification of DILI-risk compounds remains a challenge in drug discovery[13,14], emphasizing the need for alternative in vitro strategies to assess hepatotoxicity compounds and generate reliable data for CV model development. The aim of this work is to develop an expeditious tool based on a CV model for preclinical even clinical drug safety assessment.

[1]Key Laboratory of Environmental Medicine Engineering of Ministry of Education, School of Public Health, Southeast University, Nanjing, China. [2]Jiangsu Institute for Sport and Health (JISH), Nanjing, China. [3]Hepatobiliary Center of The First Affiliated Hospital, Nanjing Medical University; Research Unit of Liver Transplantation and Transplant Immunology, Chinese Academy of Medical Sciences, Nanjing, China. [4]State Key Laboratory of Digital Medical Engineering, School of Biological Science and Medical Engineering, Southeast University, Nanjing, China. [5]These authors contributed equally: Shiyi Tan, Yan Ding.
✉e-mail: zhangjuan@seu.edu.cn; gu@seu.edu.cn

Organoid culture technology provided new experimentally tractable, physiologically relevant models of human pathologies and subsequent drug screening[15]. Human liver organoids (HLOs) offer distinct advantages over HepG2 spheroids, as they comprise both hepatic parenchymal and non-parenchymal cells, reflecting accurate intercellular interactions. As 3D multicellular clusters, HLOs carry a cytochrome P450 system involved in drug metabolism, and preserve the phenotype and function of hepatocytes longer than primary human hepatocytes (PHHs)[16,17]. Recent studies highlight the potential of artificial intelligence (AI)-driven image processing to explore the strong correlation between organoid morphology and compound toxicity or disease status[18,19]. Therefore, the organoid model is not only a viable alternative to the animal model but also a promising tool primed for assessing DILI risks through morphological analysis. In image analysis, dyes are frequently used to highlight cell features, and CV techniques are then utilized to identify any changes[20]. Notably, brightfield imaging surpasses fluorescence imaging in several aspects: real-time capabilities, non-destructive nature, and the absence of additional sample processing requirements. Furthermore, brightfield imaging excels in information retrieval due to its high capacity, richness, and depth, all while being cost- and time-effective. To capture the 3D features of the organoid model, we applied 3D video processing principles and developed a CV model based on image-spatial-temporal coding layers to extract spatio-temporal information from high-content screening (HCS). Herein, we developed an evaluation system, named DILITracer, capable of predicting the clinical DILI of compounds based on the HLO technology platform and an AI-assisted algorithm for data analysis. The model achieved an impressive overall accuracy of 82.34%, with particularly high accuracy (90.16%) in identifying non-DILI compounds.

To our knowledge, DILITracer is the first model able to categorize hepatotoxicity levels (no, less, or most DILI levels) rather than merely dictating hepatotoxicity. It is simple, non-destructive, and low-cost, with rich information extracted, making it ideal for high-throughput DILI risk evaluation. Our endeavor also represents a significant advancement in compliance with the principles of the 3Rs (Replacement, Reduction, and Refinement). In summary, our innovative AI model utilizes clinical data as an endpoint categorization label, providing a rapid and simple approach to accurately screen compounds with potential clinical liver injury effects.

## Results
### The strategy for the DILI-level evaluation system based on the morphology of HLO under brightfield

As shown in Fig. 1, our approach consists of two stages: system construction and system application: (1) In the stage of system construction: we ensured that the HLOs were in a "drug-ready" state. We also selected 30 structurally and functionally representative compounds with known levels of DILI from DILIrank database[21], including four pairs of toxic drugs and their non-toxic structural analogs (troglitazone & pioglitazone, tolcapone & entacapone, nefazodone & buspirone, trovafloxacin & levofloxacin), as well as 22 drugs known to cover known DILI mechanisms[22,23] (such as mitochondrial injury, reactive metabolites, biliary transport inhibition, and immune responses) for drug testing. It is noteworthy that the Food and Drug Administration (FDA) DILIrank database categorizes compounds into different degrees of hepatotoxicity based on clinical data, confirming the close alignment of our model with clinical reality during the prediction process. Throughout the testing process, we continuously collected brightfield images of the dosed HLOs using a HCS imager. We then used the DILI levels (No, Less, and Most) of the tested compounds as labels that were added to a series of brightfield images of the corresponding HLOs to generate image sequence DILI-level data pairs. Finally, we trained an AI model to predict the DILI level based on the image sequences to learn the relationship between the image sequences and the DILI severity; (2) In the stage of system application: HLOs, also in the "drug-ready" state, were activated by compounds with unknown DILI severity and then HLO brightfield images were continuously collected during the testing process. The corresponding sequence of brightfield images was input into the AI model to obtain the predicted value

of DILI level. The model of this work exhibited an overall accuracy of 82.34%, with a particularly impressive performance in the vNo-DILI-concern category, where it achieved an accuracy of 90.16% (Table 1). This highlights the model's exceptional ability to identify compounds with no DILI risk, ensuring a high degree of reliability in distinguishing non-hepatotoxic compounds. More detailed comparisons will be discussed in the subsequent sections.

### Stable establishment of the DILI toxicity testing platform using two distinct 3D liver models

To explore the suitability of liver models with varying levels of complexity for DILI toxicity testing platforms, we established a single-type cell 3D model (HepG2 spheroid) and a multi-type cell 3D model (liver organoid). Specifically, HepG2 spheroids and HLOs were each exposed to 30 compounds with or without hepatotoxicity. The levels of ALB from the supernatant and cellular activity (ATP) from the spheres were further assessed at the end of Day 3 to validate the reliability of the system (Figs. 2b, d, f and 3b, d, f). For detailed results regarding changes in ALB and ATP levels of two in vitro 3D models under the treatment of 30 compounds, please refer to Supplementary Figs. 1–4. The brightfield images across different time series and different z-axis orientations were collected daily to generate image data for morphological analysis (Supplementary Figs. 5 and 6). Taking the HLO-based DILI toxicity platform as an example, the significant difference by compounds classified at different levels of liver toxicity potency could be observed. When treated with chlorpheniramine, labeled "No-DILI" by DILIrank, HLOs still increased in diameter and developed into a typical translucent hollow sphere with clear boundaries. In contrast to non-hepatotoxic compounds, Gefitinib-stimulated HLOs, labeled as "Most-DILI", underwent cell death, failed to maintain their original spherical structure, and disintegrated by the end of Day 3. The state of HLOs treated with Simvastatin (with a label of "Less-DILI") was between No- and Most-DILI, i.e., HLOs showed growth inhibition but their morphology was still in the form of a complete sphere. Overall, we provided a robust biological basis for the subsequent development of DILI risk prediction models (Fig. 2a, c, e).

### Comparison of DILI-level model using two distinct 3D liver models

On this basis, we comprehensively compared the performance of the image-only model across two different experimental platforms (Table 1). As shown in Fig. 4a–d, the DILI classifier exhibited a commendable predictive performance on the dataset of the HLOs platform, with an accuracy of 82.34%, far exceeded by HepG2 spheroids (an average accuracy of 77.41%). Notably, the HLO-based model correctly identified 90.16% of the actual cases of vNo-DILI-concern. Furthermore, for the vNo- and vLess-DILI-Concern cases, the recall of the model for the HLO dataset exceeded those for the HepG2 spheroids, signaling the remarkably higher predictive power of the model when utilizing the HLOs platform. Also, the HLO-based model exhibited robust specificity and had a better capability than HepG2 spheroid in effectively identifying instances that did not pertain to compounds belonging to vLess- and vMost-DILI-Concern. The notion that organoid models outperform HepG2 spheroid models is further substantiated by their superior precision.

However, some limitations still exist in the model using the HLO dataset. The model using the HLO dataset lacked accuracy in labeling the true positives among all the actual vMost-DILI-Concern samples compared to the HepG2 spheroid. Another caveat of the model using the HLO dataset was the lower likelihood of labeling correctly samples as non-vNo-DILI-Concern than using HepG2 spheroid (85.57% vs. 94.16%). As an indicator of balance in precision and recall, the F1 score was slightly lower for HLOs in the vMost-DILI-Concern classification. This reflected the previously mentioned lower recall, indicating a minor imbalance in the model that favored minimizing false positives at the expense of potentially missing true positives. The AUC value of vNo-, vLess-, and vMost-DILI-Concern in our prediction model using HLOs and HepG2 spheroids has been shown in Fig. 4e, f, indicating the reliable performance of our prediction models in

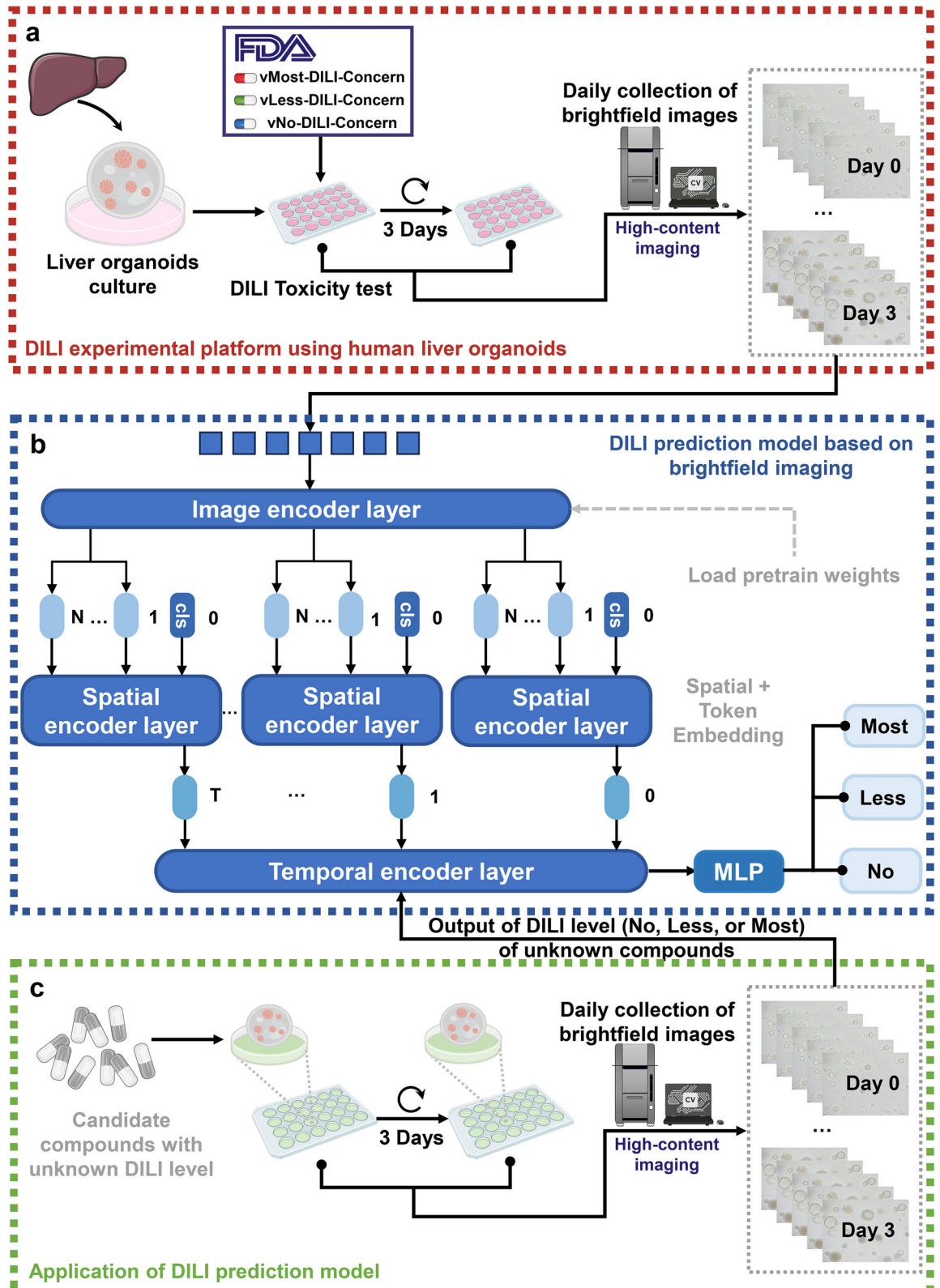

**Fig. 1 | The workflow of the development of the DILI-level prediction model.**
**a** The construction of the DILI experimental platform based on human liver organoids for collecting daily brightfield images treated with different compounds under different *z*-axis. **b** The establishment of DILI-level prediction model using different brightfield images of liver organoids with spatiotemporal information. **c** The application of AI model for predicting DILI level of compounds with unknown hepatotoxicity.

**Table 1 | Predictive performance metrics comparison between two in vivo 3D platforms**

| Metrics | | Human liver organoids | HepG2 spheroids |
|---|---|---|---|
| Recall | vNo-DILI-Concern | 0.9016 | 0.5800 |
| | vLess-DILI-Concern | 0.7500 | 0.6058 |
| | vMost-DILI-Concern | 0.7808 | 0.8580 |
| Specificity | vNo-DILI-Concern | 0.8557 | 0.9416 |
| | vLess-DILI-Concern | 0.9552 | 0.9243 |
| | vMost-DILI-Concern | 0.9059 | 0.6429 |
| Precision | vNo-DILI-Concern | 0.7971 | 0.5370 |
| | vLess-DILI-Concern | 0.7500 | 0.6923 |
| | vMost-DILI-Concern | 0.8769 | 0.6429 |
| F1 score | vNo-DILI-Concern | 0.8462 | 0.5577 |
| | vLess-DILI-Concern | 0.7500 | 0.6462 |
| | vMost-DILI-Concern | 0.8261 | 0.8463 |
| Accuracy | | 0.8234 | 0.7741 |

classifying different labels. Overall, judging by the five model evaluation criteria, the DILI-classification model performed relatively better when using HLOs compared to HepG2 spheroids. Also, the findings revealed a commendable discriminatory capability of our HLO-based model in distinguishing instances of vNo-DILI-Concern from others.

## Superiority analysis of the DILI prediction model using HLOs from in vitro and in silico perspectives

Next, we attempted to demonstrate the superiority of our AI model from the perspective of in vitro biological models. The result of immunofluorescence (Fig. 5a) confirmed liver-specific "bile duct-like structure" as indicated by the markers of bile salt export pump (BSEP). Also, tight junction protein stained by zonula occludens-1 (ZO-1) suggested a multi-cellular-type 3D hollow body, including hepatocytes rich with hepatocyte nuclear factor 4-alpha (HNF4a), CD31-expressing liver sinusoidal endothelial cells, CD68$^+$ Kupffer cells, and DES-containing hepatic stellate cells. Importantly, the HLO model showed significantly higher expression levels of metabolic enzyme *CYP34A* (1.80 folds), *CYP1A2* (88.96 folds), *CYP2D6* (4.95 folds), *CYP2E1* (10.79 folds), *CYP2C9* (8.16 folds), and *CYP2C19* (8.62 folds) than those in HepG2 spheroids (Fig. 5b). Therefore, we assumed that liver organoids, as a more physiologically relevant in vitro liver model, would be able to generate more realistic toxicological responses and thus provide more reliable image data for the development of DILI models.

We further conducted ablation experiments to investigate the impact of temporal and spatial modalities on the DILI prediction model (Table 2). First, we used Day 0–Day 1 image data instead of Day 0–Day 3 images to evaluate the role of temporal dimension information. The results showed a 12.09% decrease in prediction accuracy compared to the original model, with lower recall, specificity, precision, and F1 scores for each label (Fig. 5c). Second, we replaced multiple separate images of the same sample taken at different heights with 3D composite images generated by a fusion algorithm (provided by the HCS instrument), which combines images from different focal planes. After removing the spatial coding layer from the model, the prediction accuracy dropped by 6.24% compared to the original model. Recall, precision, and F1 scores for each label did not surpass the original model's metrics (Fig. 5d). Overall, both temporal and spatial modalities exerted a substantial positive influence on the development of DILI prediction models, thus not only validating the soundness and effectiveness of the modeling approach but also emphasizing the pivotal role of temporal and spatial dimensions in replicating intricate biological mechanisms.

## Attention mechanism visualization for DILI prediction model based on HLOs

To further verify that the model effectively learned the morphological features of HLOs before and after drug exposure, we visualized the model's

output. Typical samples belonging to the Most- and No-DILI categories were selected to input into the STViT model, and then output attentional heat maps based on the attentional weights of the corresponding samples to determine the importance of each part of the HLO images in the model decision. Figure 5e, f were two sets of images (the overlay of the heatmap and the brightfield image) at different z-axis after drug exposure from Day 0 to Day 3. We found that on the same z-axis, the high attentional weights were predominantly concentrated on the HLOs that were accurately focused within the current visual field. Meanwhile, attentional weights grew significantly over time in response to significant changes in organoid activity (e.g., disintegration or growth). For example, in the Gefitinib-treated positive samples, the HLOs in the area marked by the small red box (Day 2) started to undergo significant cell death and structural disintegration. And in the Chlorphenamine-treated negative samples, the HLOs (Day 2 or Day 3) in the marked area still grew significantly. Correspondingly, both of these areas were assigned higher attentional weights by the model. The results above indicated that the model can accurately locate the position of HLOs and understand morphologically the change of organoid status.

## Discussion

In this study, we successfully developed a DILI prediction model based on organoids, which we named "DILITracer" to highlight its ability to "trace" the DILI level (Most-, Less-, or No-DILI). Our model achieved an average accuracy of 82.34%, demonstrating improved predictive performance for DILI prediction compared to HepG2 spheroids and animal models. Almost all of the indicators (recall, specificity, precision, and F1 score) of each classification label exhibited a better value in the prediction model using HLO imaging compared to HepG2 spheroids. Our organoid experimental platform has been demonstrated to effectively mimic cell-cell interactions and exhibit higher levels of functional cytochrome P450 enzymes, suggesting that organoids serve as a more physiologically relevant in vitro 3D liver model compared to HepG2 spheroids. Furthermore, the generation of a comprehensive series of image data capturing detailed morphological features of organoids could provide a convenient and effective approach to reflect more realistic toxicological responses, thereby facilitating the establishment of robust AI models. Importantly, our model has the potential to identify certain "clinically specific toxic drugs" that induce liver toxicity clinically, despite having passed standard preclinical toxicology evaluations using animal models prior to first-in-human administration. Specifically, our model successfully identified simvastatin and stavudine as "non-No-DILI" cases, which had been poorly predicted by hepatic spheroids in a previous study[24]. This may be partly attributed to the clinical relevance of the labels used in our model, where we employed clinical data-based drug classifications from the FDA DILIrank database for model training. This approach ensures that our model is closely aligned with clinical reality,

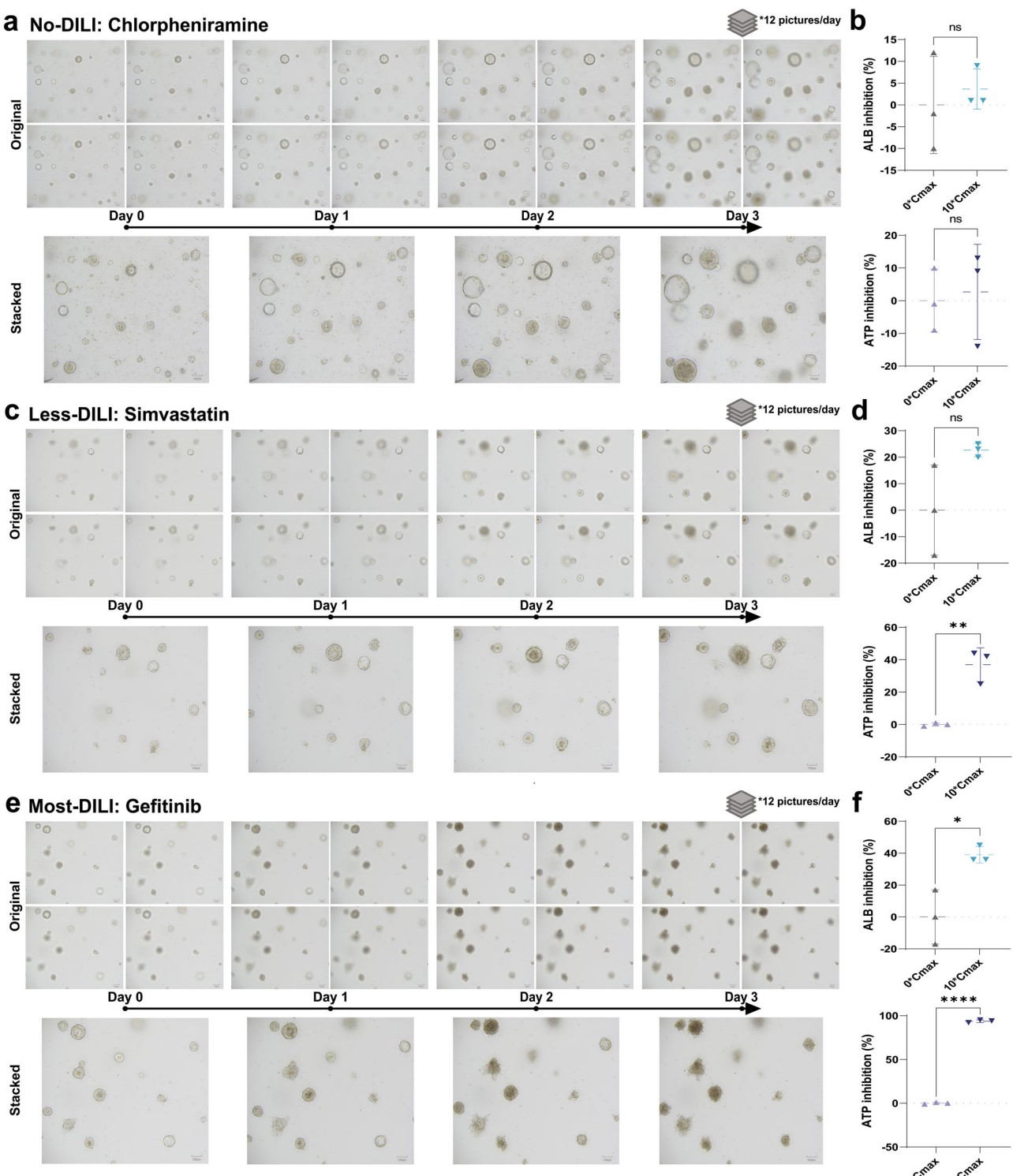

**Fig. 2 | DILI toxicity testing platform based on human liver organoid models.**
**a** Brightfield morphology changes (12 original images and 1 stacked image) of liver injury in human liver organoids exposed to Chlorpheniramine (with a label of No-DILI levels) from Day 0 to Day 3. **b** ALB inhibition ($P = 0.6262$, $t = 0.5268$) and ATP inhibition ($P = 0.8040$, $t = 0.2652$) of liver injury in human liver organoids exposed to Chlorpheniramine at end of Day 3. **c** Brightfield morphology changes (12 original images and 1 stacked image) of liver injury in human liver organoids liver organoids exposed to Simvastatin (with a label of Less-DILI levels) from Day 0 to Day 3. **d** ALB inhibition ($P = 0.0844$, $t = 2.285$) and ATP inhibition ($P = 0.0036$, $t = 6.110$) of liver

injury in human liver organoids exposed to Simvastatin at end of Day 3. **e** Brightfield morphology changes (12 original images and 1 stacked image) of liver injury in human liver organoids exposed to Gefitinib (with a label of Most-DILI levels) from Day 0 to Day 3. **f** ALB inhibition ($P = 0.0191$, $t = 3.800$) and ATP inhibition ($P < 0.0001$, $t = 88.86$) of liver injury in human liver organoids exposed to Gefitinib at end of Day 3. Scale bar = 100 μm. *$P < 0.05$; **$P < 0.01$; ****$P < 0.0001$. Error bars represent mean ± SD across $n = 3$ biologically independent organoid samples. Effect sizes (Cohen's $d$) and 95% confidence intervals (CIs) are provided in Supplementary Table 5.

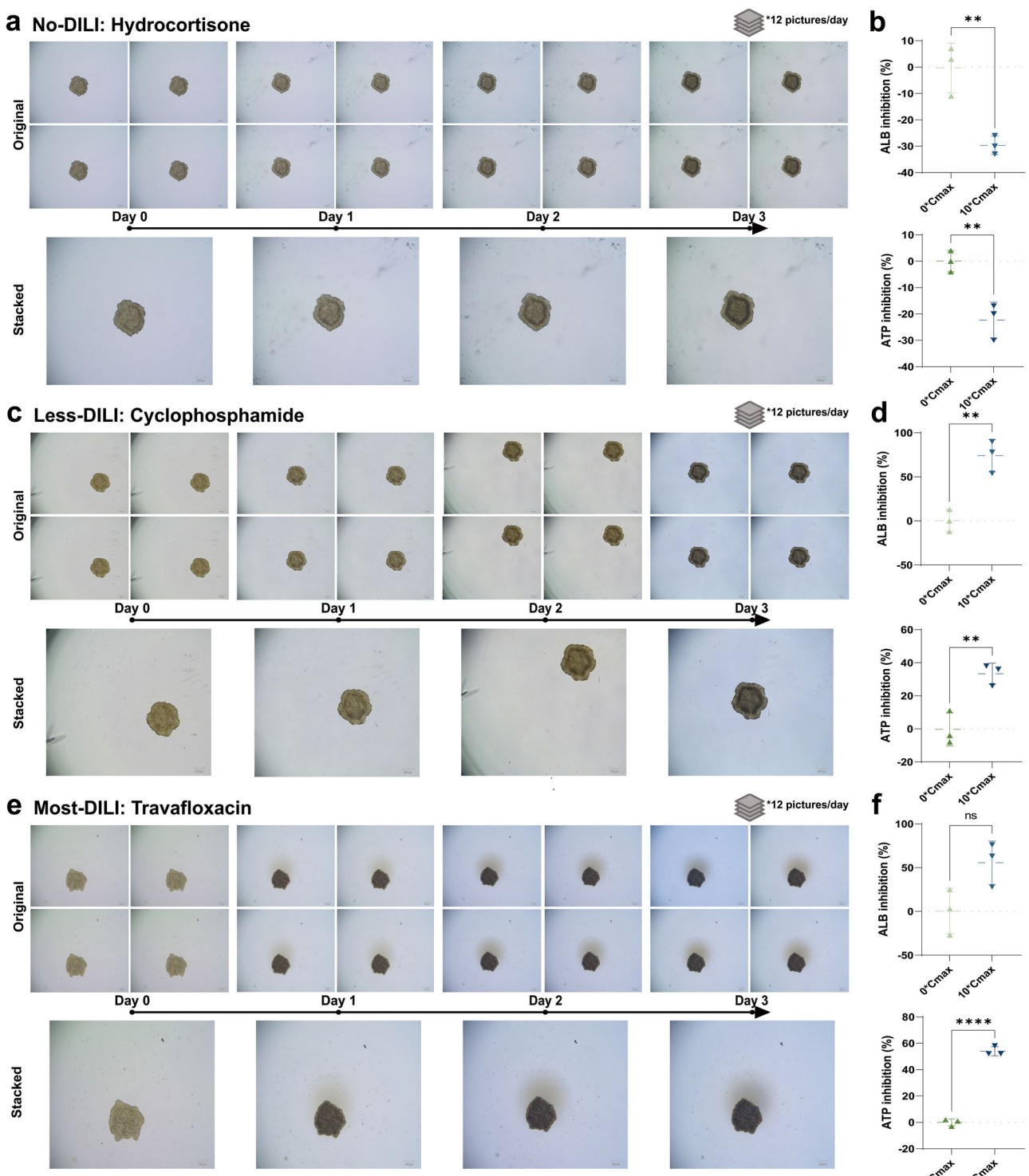

**Fig. 3 | DILI toxicity testing platform based on HepG2 spheroid models.**
a Brightfield morphology changes (12 original images and 1 stacked image) of liver
injury in HepG2 spheroids exposed to Hydrocortisone (with a label of No-DILI
levels) from Day 0 to Day 3. b ALB inhibition ($P = 0.0073$, $t = 5.039$) and ATP
inhibition ($P = 0.0080$, $t = 4.900$) of liver injury in HepG2 spheroids exposed to
Hydrocortisone at end of Day 3. c Brightfield morphology changes (12 original
images and 1 stacked image) of liver injury in HepG2 spheroids exposed to Cyclo-
phosphamide (with a label of Less-DILI levels) from Day 0 to Day 3. d ALB inhi-
bition ($P = 0.0045$, $t = 5.750$) and ATP inhibition ($P = 0.0080$, $t = 4.899$) of liver

injury in HepG2 spheroids exposed to Cyclophosphamide at end of Day 3.
e Brightfield morphology changes (12 original images and 1 stacked image) of liver
injury in HepG2 spheroids exposed to Travafloxacin (with a label of Most-DILI
levels) from Day 0 to Day 3. f ALB inhibition ($P = 0.0564$, $t = 2.661$) and ATP
inhibition ($P < 0.0001$, $t = 21.46$) of liver injury in HepG2 spheroids exposed to
Travafloxacin at end of Day 3. Scale bar = 100 μm. **$P < 0.01$; ****$P < 0.0001$. Error
bars represent mean ± SD across $n = 3$ biologically independent organoid samples.
Cohen's $d$ and 95% CI are provided in Supplementary Table 6.

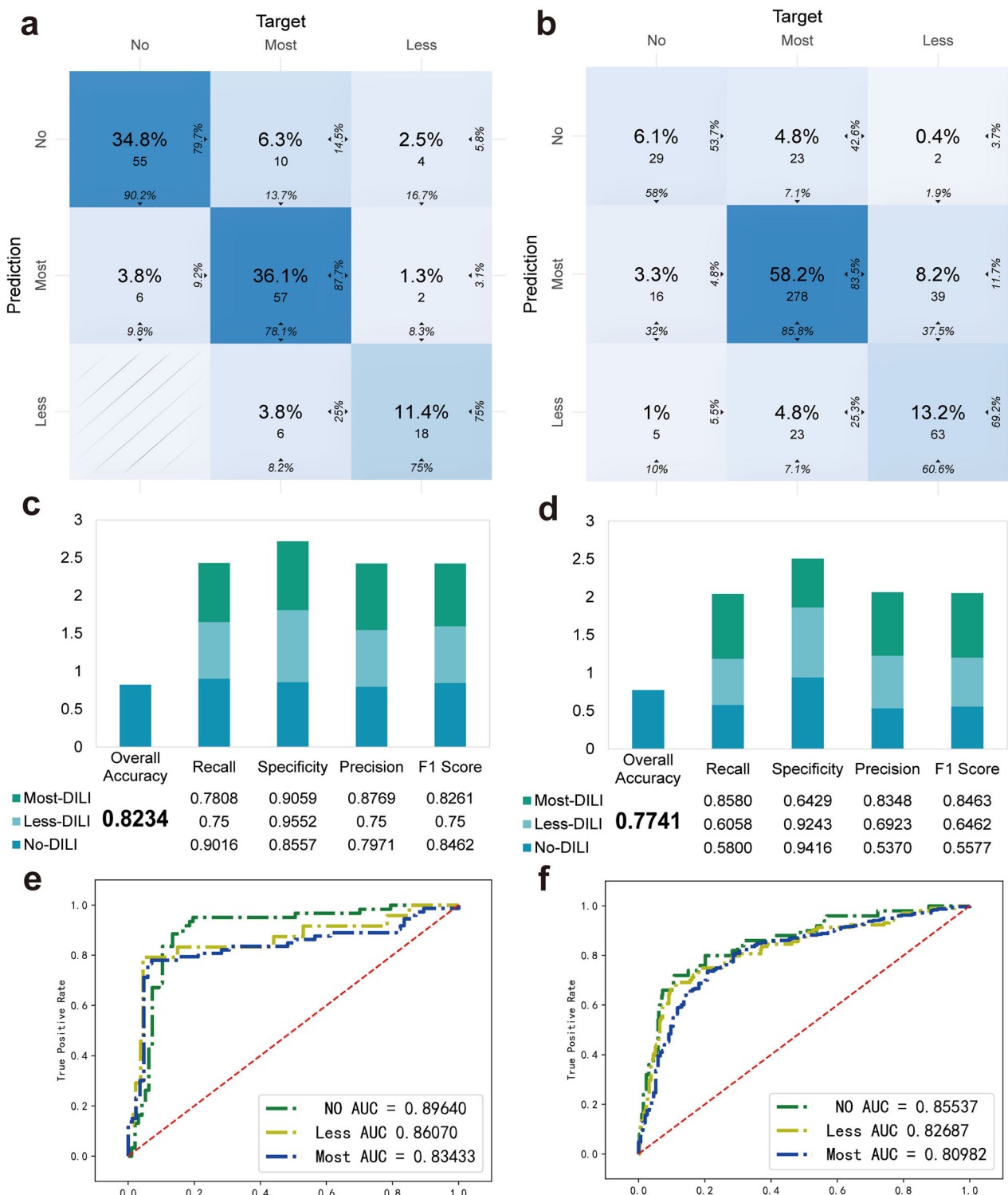

**Fig. 4 | Evaluation of the predictive performance of human liver organoids and HepG2 spheroids image-only models. a, b** Confusion matrix of DILI classifier based on the human liver organoid and HepG2 spheroid dataset. **c, d** Indexes of overall accuracy, recall, specificity, precision, and F1 score of DILI classifier based on the human liver organoid and HepG2 spheroid dataset. **e, f** Receiver operating characteristic (ROC) curves of the human liver organoid and HepG2 spheroid image-only model. Green, yellow, and blue curves represent vNo-, vLess-, and vMost-DILI-Concern labels, respectively. AUC values are labeled for each curve.

highlighting its significant practical value in clinical drug development and safety assessment. Overall, we believe it qualifies as a suitable tool for predicting DILI during preclinical drug development.

A previous study has developed a DILI prediction strategy based on features of fluorescence images of PHHs analyzed by a random forest algorithm[25]. However, the fluorescent dye or probe, as an invasive way, was limited to detection at endpoints of toxicity. Therefore, we collected non-destructive brightfield images across different time series (once a day for a total of 4 days) to realize the "dynamic monitoring" when clarifying the DILI toxicity. Currently, describing the structure of organoids with high phenotypic complexity using traditional morphological features such as radius length, area, and perimeter is challenging. Deep learning, however, offers a

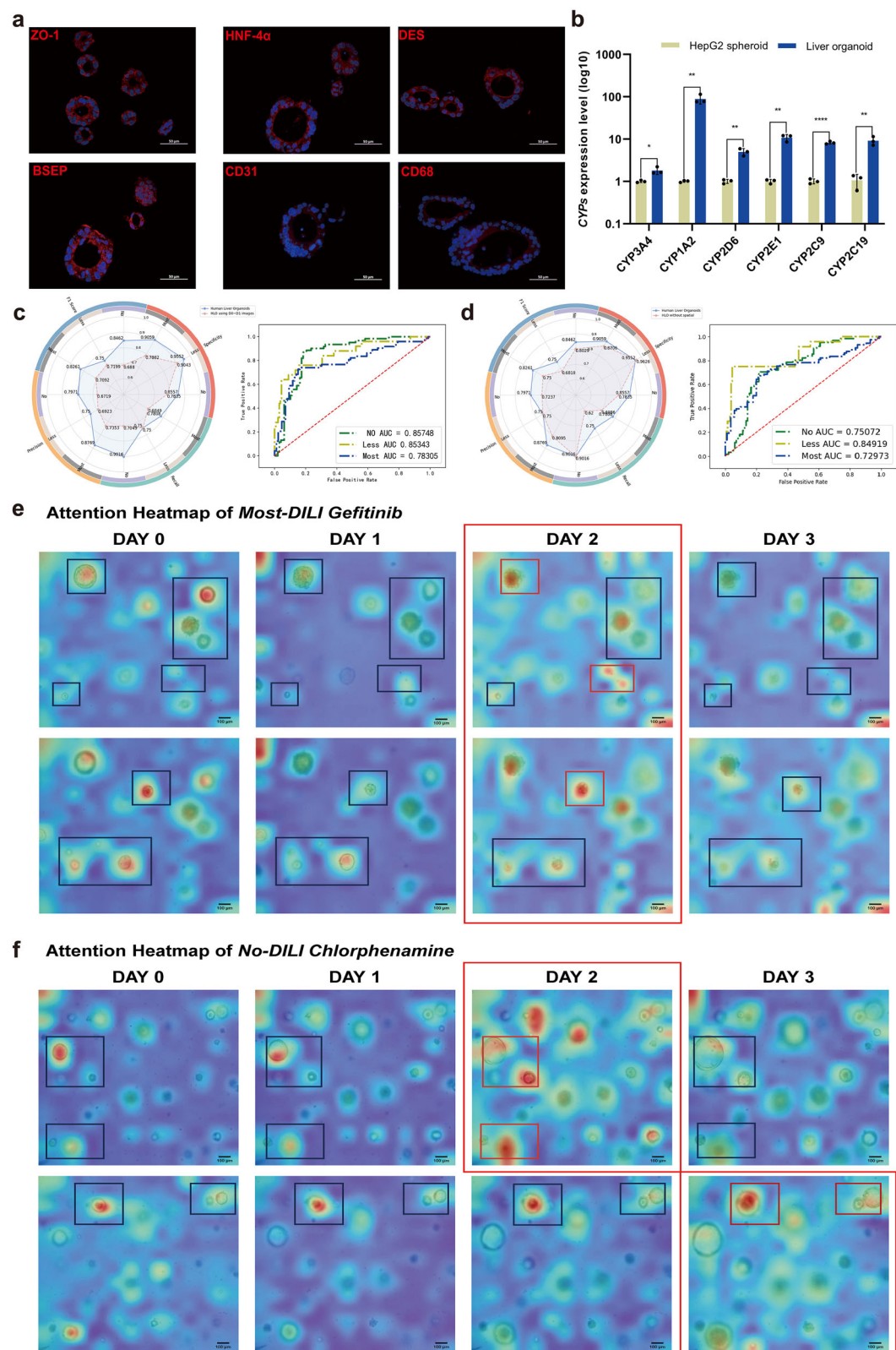

viable solution by effectively capturing the intricate patterns and features of organoids[19,26,27]. To date, for the construction of the DILI prediction model, neither 2D nor 3D imaging technologies have been used in combination with CV techniques based on deep learning. In this study, we referred to the technical principles of video processing to fully excavate the spatial and temporal features of brightfield images. These two features are extremely

important for us to generate a DILI image-only model with a favorable predictive performance, as evidenced by ablation experiments: the accuracy of our establishing model (82.34%) is extremely higher than that of the model without spatial feature (76.10%) and model without temporal feature (70.25%). Also, we categorized input labels into three groups of DILIranks (vMost-, vLess-, and vNo-DILI concern) based on confirmed causal

**Fig. 5 | Superiority analysis and ability validation of human liver organoids image-only model. a** Immunofluorescence detection of ZO-1, BSEP, HNF4a, DES, CD31, CD68 in human liver organoids. Scale bar = 50 μm. **b** Expression level of *CYP3A4* ($P = 0.0175$, $t = 3.900$), *CYP1A2* ($P = 0.0022$, $t = 7.022$), *CYP2D6* ($P = 0.0021$, $t = 7.108$), *CYP2E1* ($P = 0.0010$, $t = 8.505$), *CYP2C9* ($P < 0.0001$, $t = 20.51$), and *CYP2C19* ($P = 0.0031$, $t = 6.374$) between HepG2 spheroids and human liver organoids. *$P < 0.05$, ** $P < 0.01$; ****$P < 0.0001$. Error bars represent mean ± SD across $n = 3$ biologically independent organoid samples. Cohen's *d* and 95% CI are provided in Supplementary Table 7. **c** Radar plot visualizing the confusion matrix in ablation experiments based on spatial dimensions and receiver operating characteristic (ROC) curves of human liver organoids image only model

using Day0–Day1 images. **d** Radar plot visualizing the confusion matrix in ablation experiments based on temporal dimensions and ROC curves of human liver organoids image only model without the structure of the spatial coding layer. Green, yellow, and blue curves represent vNo-, vLess-, and vMost-DILI-Concern labels, respectively. **e** Attention heatmap of Gefitinib. *z*-axis (top image) = 135,650; *z*-axis (bottom image) = 135,950; **f** attention heatmap of Chlorphenamine. *z*-axis (top image) = 136,500; *z*-axis (bottom image) = 136,900; the red and the blue areas in the heatmap indicate the areas with higher and lower attention weights, respectively. The small boxes mark the areas that are in focus precisely under each visual field. The large red boxes mark the time period with higher activation of attention weights. Scale bar = 100 μm.

### Table 2 | Predictive performance metrics in ablation experiments

| Metrics | | HLO without spatial | HLO using D0–D1 images | HLO |
|---|---|---|---|---|
| Recall | vNo-DILI-Concern | 0.9016 | 0.7049 | 0.9016 |
| | vLess-DILI-Concern | 0.6200 | 0.7500 | 0.7500 |
| | vMost-DILI-Concern | 0.6986 | 0.6849 | 0.7808 |
| Specificity | vNo-DILI-Concern | 0.7835 | 0.7835 | 0.8557 |
| | vLess-DILI-Concern | 0.9626 | 0.9043 | 0.9552 |
| | vMost-DILI-Concern | 0.8706 | 0.7882 | 0.9059 |
| Precision | vNo-DILI-Concern | 0.7237 | 0.6719 | 0.7971 |
| | vLess-DILI-Concern | 0.7500 | 0.6923 | 0.7500 |
| | vMost-DILI-Concern | 0.8095 | 0.7353 | 0.8769 |
| F1 score | vNo-DILI-Concern | 0.8029 | 0.6880 | 0.8462 |
| | vLess-DILI-Concern | 0.6818 | 0.7199 | 0.7500 |
| | vMost-DILI-Concern | 0.7500 | 0.7092 | 0.8261 |
| Accuracy | | 0.7610 | 0.7025 | 0.8234 |

evidence in clinical linking a drug to liver injury, providing a more nuanced assessment of hepatotoxicity. To the best of our knowledge, this is the first model to output ternary classification of hepatotoxicity rather than simply indicating whether or not hepatotoxicity is present.

The integration of our HLO-based DILI prediction model into preclinical testing workflows has the potential to revolutionize drug safety assessment. By providing an early-stage, in vitro platform for hepatotoxicity evaluation, our model might significantly reduce reliance on animal models, which often struggle to predict DILI, particularly idiosyncratic DILI[28]. Compared to previous DILI prediction models that rely on chemical structure[29–31] or gene expression[32,33] as data modalities, our approach offers a more convenient data acquisition process. Moreover, the early identification of clinically relevant toxic drugs during preclinical testing enables the detection of compounds with a high risk of liver toxicity before they advance to human trials, ultimately reducing costly late-stage failures. In this study, our model's high accuracy in identifying vNo-DILI cases (90.16%) ensures that safe drugs are prioritized for clinical trials, minimizing DILI risk and improving the likelihood of success in clinical trials and new drug projects. This approach may help lower drug development costs, provide further insights into liver toxicity risks, and offer a more reliable reference for clinical decision-making. Interestingly, the attention mechanism employed in this study revealed that our model is capable of identifying critical time points for distinguishing drug effects on organoids. This might provide valuable insights into the time window of clinical toxicity efficacy, serving as an important reference for optimizing clinical monitoring and intervention strategies—an area that warrants further investigation in future studies. By integrating dynamic brightfield imaging, machine learning, and clinical data from the FDA DILIrank database, our model offers an opportunity to enhance the predictive reliability of early-stage toxicity screening. Additionally, its non-invasive nature and real-time monitoring capabilities can be seamlessly incorporated into existing drug safety pipelines, facilitating more efficient drug development and the early

elimination of hepatotoxic compounds. Overall, we hope to accelerate the transition of the DILI prediction model using organoids "from the bench to the bedside."

There are still some limitations. The imbalance among the vMost-, -vLess, and vNo-DILI-Concern sample sizes may also contribute to the significant discrepancy of accuracy across different categories. Also, HLOs remain simplified and lack immune components in comparison to organ-on-a-chip[10], potentially leading to false-negative results for certain compounds. In the future, we will focus on additional modifications of organoid or organoid-on-a-chip platforms to fully predict DILIs of different mechanisms, particularly immune-mediated DILI.

In this study, we successfully developed DILITracer, a DILI prediction model that analyzes spatiotemporal features from continuously captured brightfield images of liver organoids under various DILI conditions. The model correlates organoid morphology with DILI severity, providing risk assessments for compounds categorized as most-, less-, or no-DILI. DILITracer demonstrates impressive accuracy in predicting DILI levels and incorporates clinical data as outcome variables, ensuring strong clinical relevance. This AI-driven system offers a rapid and reliable tool for predicting hepatotoxicity in early-stage drug development and provides valuable insights for clinical drug screening.

## Methods
### Culture of HLOs and HepG2 spheroids
HLOs from liver cancer patient adjacent normal tissue were donated by Avatarget Co., Ltd (Suzhou, China), under informed consent and ethical approval (the Ethics Committee of the First Affiliated Hospital of Nanjing Medical University, 2021-SR-575). HLOs were embedded in Matrigel (Corning, 356231) and cultured in corresponding media (Avatarget, KLV0010101). All experiments were performed using HLOs derived from a single donor tissue. For each assay, three technical replicates were analyzed, corresponding to three independently cultured wells seeded with organoids. For passage, the following embedding method was used: HLOs were

**Table 3 | List of drugs tested in HepG2 spheroids or liver organoids**

| No. | Compounds | DILIrank | Severity class | Label section |
|---|---|---|---|---|
| 1 | Troglitazone | vMost-DILI-Concern | 8 | Withdrawn |
| 2 | Pioglitazone | vLess-DILI-Concern | 3 | Warnings and precautions |
| 3 | Tolcapone | vMost-DILI-Concern | 8 | Box warning (withdrawn) |
| 4 | Entacapone | vLess-DILI-Concern | 0 | No match |
| 5 | Nefazodone | vMost-DILI-Concern | 8 | Box warning |
| 6 | Buspirone | Ambiguous DILI-concern | 3 | Adverse reactions |
| 7 | Trovafloxacin | vMost-DILI-Concern | 8 | Withdrawn |
| 8 | Levofloxacin | vMost-DILI-Concern | 8 | Warnings and precautions |
| 9 | Cyclophosphamide | vLess-DILI-Concern | 5 | Adverse reactions |
| 10 | Methotrexate | vMost-DILI-Concern | 3 | Box warning |
| 11 | Paclitaxel | vLess-DILI-Concern | 4 | Adverse reactions |
| 12 | Erlotinib | vMost-DILI-Concern | 8 | Warnings and precautions |
| 13 | Gefitinib | vMost-DILI-Concern | 4 | Warnings and precautions |
| 14 | Dasatinib | vLess-DILI-Concern | 4 | Adverse reactions |
| 15 | Vinblastine | vNo-DILI-Concern | 0 | No match |
| 16 | Acetaminophen | vMost-DILI-Concern | 5 | Warnings and precautions |
| 17 | Benzbromarone | vMost-DILI-Concern | 8 | Withdrawn |
| 18 | Bosentan | vMost-DILI-Concern | 7 | Box warning |
| 19 | Diclofenac | vMost-DILI-Concern | 8 | Warnings and precautions |
| 20 | Flutamine | vMost-DILI-Concern | 8 | Box warning |
| 21 | Isoniazid | vMost-DILI-Concern | 8 | Box warning |
| 22 | Nimesulide | vMost-DILI-Concern | 8 | Withdrawn |
| 23 | Stavudine | vMost-DILI-Concern | 8 | Box warning |
| 24 | Simvastatin | vLess-DILI-Concern | 3 | Warnings and precautions |
| 25 | Zileuton | vMost-DILI-Concern | 5 | Warnings and precautions |
| 26 | Chlorpheniramine | vNo-DILI-Concern | 0 | No match |
| 27 | Digoxin | vNo-DILI-Concern | 0 | No match |
| 28 | Diphenhydramine | vNo-DILI-Concern | 0 | No match |
| 29 | Hydrocortisone | vNo-DILI-Concern | 0 | No match |
| 30 | Lidocaine | vNo-DILI-Concern | 0 | No match |

enzymatically or mechanically fragmented (with pre-cold phosphate-buffered saline (PBS)) and reseeded in new Matrigel. Specifically, the pre-cold Matrigel/cell mixture is seeded in 24-well plates at 30 μL/well to enable the formation of dome-shaped structures. Incubation in the cell incubator (37 °C, 5% $CO_2$) for 15 min is required for Matrigel polymerization. After solidification, 500 μL of specific medium is added and later renewed at specific intervals.

The HepG2 cells were donated by Avatarget Co., Ltd (Suzhou, China). The HepG2 cells were cultured in DMEM (Gibco, 10567014) containing 10% fetal bovine serum (GIBCO, A3161001C), and 1×penicillin/streptomycin (Gibco, 15140148). HepG2 cells (70–80% cell confluency) were washed twice in sterile PBS and then dissociated with 1 mL 0.25% EDTA-Trypsin. Using 3×volume of the complete medium neutralized with reagent above, count cells and ensure 10,000 cells/well. The desired volume of cell suspension was inoculated into the 200 μL/well of ultra-low attachment microplate (Corning, 7007). The plate was centrifuged at 1500 rpm for 10 min and then incubated in an incubator at 37 °C in 5% $CO_2$ for 4 days. The medium is changed every other day at a 1:1 ratio.

#### Compounds screening
According to the DILI classification by DILIrank[21], the Liver microphysiological systems development guidelines[22], we selected 30 compounds for testing. All selected compounds underwent extensive Liver-Chip-based

DILI testing according to Emulate Co., Ltd[34]. The specific list of DILI-related drugs is shown in Table 3.

#### DILI toxicity test
The peak serum concentration (Cmax) can intuitively reflect the actual presence level of a drug in the body's circulatory system. Therefore, we used 1*Cmax as a reference for the drug dosage to better approximate the in vivo conditions. On this basis, we selected 10*Cmax as the reference concentration, taking into account the coefficient of variation in toxicology (individual variability: tenfold). Also, based on existing literature[25,34], we included 100*Cmax to account for the possibility of higher drug concentrations in extreme or therapeutic scenarios, thereby ensuring a more comprehensive assessment of drugs that may exhibit toxicity under elevated exposure conditions. In summary, for HepG2 spheroids, the concentration gradients of 0, 1, 10, and 100*Cmax were selected to cover a range of drug exposure scenarios. Each drug was tested in triplicate. Plates were dosed with the drug for 72 h in each cycle (referred to as Day 0 through Day 3). The image dataset for HepG2 spheroid-based DILI test was further trained using multimodal ViLT and iRENE models, revealing that the concentration of 10*Cmax yielded the highest accuracy in the average test set (Supplementary Tables 1 and 2). Consequently, for the subsequent DILI assay on HLOs, only the concentration of 10*Cmax was used. At the end of day 3, the supernatant from each well was collected to determine ALB secretion, and

the spheroids from each well were used for the cell viability assay. The Cmax information of the drugs is shown in Supplementary Table 3.

### Albumin, urea, and alanine transaminase assay
The culture supernatants were collected and stored at −80 °C until use. The supernatant was assayed with a Human Albumin ELISA Kit (PROTEINTECH, KE00076). Assays were performed according to the manufacturer's instructions.

### Cellular viability determination
CellTiter-Glo (CTG, G9681, PROMEGA) reagent and culture medium were added to each well in a 1:1 ratio. The contents were mixed for 2 min to induce cell lysis. The plate was incubated at room temperature for 10 min to stabilize the luminescent signal. Recording of luminescence was carried out with BioTek Microplate Reader Synergy H1 (Vermont, USA).

### RNA isolation, reverse transcription (RT), and RT–quantitative polymerase chain reaction (qPCR)
Total RNA was extracted from the HepG2 spheroids or HLOs using RNA-easy Isolation Reagent (YAZYME, R701-01) according to the manufacturer's protocol. Reverse transcription was carried out using the HiScript III Reverse Transcriptase (YAZYME, R302-01) according to the manufacturer's protocol. qPCR was conducted using the SYBR Green Master Mix Kit (YAZYME, Q331-02) on a QuantStudio$^{TM}$ 5 Real-Time PCR Instrument (THERMOFISHER). The data were normalized to GAPDH as the endogenous control. Relative expression was calculated using the $2^{-\Delta\Delta Ct}$ method. The specific primer sequences are shown in Supplementary Table 4.

### Immunofluorescence
The HLOs were collected from Matrigel and fixed with 4% paraformaldehyde for 30 min, permeabilized with PBS containing 0.2% Triton X-100 (BEYOTIME, P0096) and blocked with PBS containing 5% bovine serum albumin (BIOSHARE, BS114) for 30 min at room temperature. HLOs were then incubated with primary antibodies overnight at 4 °C, followed by secondary antibodies for 1 h. Finally, the antifade mounting medium with DAPI (BEYOTIME, P0131) was used for nuclear staining. The dilutions for primary and secondary antibodies were: anti-ZO-1 (1:500, SERVICEBIO, GB111402-100), anti-ABCB11/BSEP (1:300, SERVICEBIO, GB113909), anti-HNF4a (1:100, SERVICEBIO, GB115549-100), anti-Desmin (1:500, SERVICEBIO, GB12088-100), anti-CD31 (1:300, SERVICEBIO, GB12064-100), anti-CD68 (1:500, SERVICEBIO, GB113150-100). Images were taken with an Olympus IX83 microscope (Tokyo, Japan).

### Statistics and reproducibility
All statistical analyses were performed using GraphPad Prism 9 software. Unpaired Student's $t$ tests (two-tailed) were used for comparisons between two groups. Data are presented as mean ± SD, and $P < 0.05$ was considered statistically significant. For each condition, experiments were conducted on biologically independent organoid samples, defined as organoids derived from the same donor but established and cultured independently across different batches and wells.

### Data collecting, labeling, and pre-processing
In all, 478 and 158 sample data were obtained from the HepG2 spheroids and HLO-based DILI assay, respectively. Each sample contains 4 days (Day 0 through Day 3) of 3D images, with 12–20 images taken equidistantly in the spatial dimension using the high-content imaging instrument (Avatarget). HLOs were imaged at ×10 magnification, while HepG2 spheroids were imaged at ×4 magnification. The training and testing datasets were split in an 80:20 ratio.

DILIrank is the largest reference drug list ranked by the risk of developing DILI in humans. The DILIrank dataset includes DILI risk classification, which is determined based on FDA-approved drug label

information and literature-reported causality assessments. It consists of four categories:

(1) vMost-DILI concern: drugs withdrawn due to DILI, or drugs with black box warnings or labels containing warnings, precautions, or descriptions of severe or moderate liver injury, as validated through causality assessment; (2) vLess-DILI concern: drugs assessed as low risk based on drug labels, confirmed through causality validation; (3) Ambiguous DILI concern: drugs evaluated as high or low risk based on drug labels but lacking sufficient evidence of causality; (4) vNo-DILI concern: drugs with no literature reports confirming their role in causing DILI.

Drug labeling serves as a critical basis for DILI risk classification and is derived from a systematic evaluation of preclinical toxicology data, clinical trials, post-marketing surveillance, and literature data. It provides essential drug safety information, including DILI risk, and is regarded as the "most reliable data source"[21]. Herein, we categorized the data into three groups based on DILIrank classification, i.e., vMost-, vLess, and vNo-DILI concern. Notably, from the perspective of practical application, we excluded "Ambiguous DILI concern" category to reduce the uncertainty caused by the lack of clear causal evidence.

The original images are resized to a uniform size of 224 × 224 pixels to align with the network model and optimize dataset utilization. Subsequently, we apply pixel normalization and standardization, resulting in input parameters $x \in R^{H \times W \times C}$ ($H = 224$, $W = 224$, $C = 3$).

### Model architecture, training, and validation
The CV classification model employed in this study consisted of four layers: (1) Image Encoder Layer, (2) Spatial Encoder Layer, (3) Temporal Encoder Layer, and (4) the Classification Layer. The architecture is inspired by the Video Vision Transformer (ViViT) model[2], which captures temporal dynamics in video data by treating it as sequences of images over time. Furthermore, we augment the image encoder layer with a spatial encoder layer to capture spatial relationships among images positioned at different levels along the $z$-axis in three-dimensional space. This multi-layered approach strikes an optimal balance between model complexity and performance.

(1) Image Encoder Layer: the BEiT-V2[35] model is used to comprehensively capture image features. We initialize this layer with weights pretrained on a dataset of 700,000 cell images, leveraging the model's ability to extract deep-dimensional features from cell images. The pre-training of the BEiT-V2 model occurs in two stages: the first stage involves Vector-quantized Knowledge Distillation (VQ-KD)[35], and the second stage trains the BEiT-V2 model itself using the visual symbols generated in the first stage. Visual symbols derived from VQ-KD serve as the training targets of the subsequent stage.

Contrastive language-image pre-training (CLIP)[36] is employed as the teacher model in the VQ-KD model of the first stage. In the second stage, for a given input image $x \in R^{H \times W \times C}$, it is reshaped to $N = HW/P^2$ patches $\{x_i^p\}_{i=1}^N$, where $x^p \in R^{N \times (P^2 C)}$ and $(P, P)$ is the patch size. In this experiment, each 224 × 224 image is divided into a 14 × 14 grid of image patches, with each patch being 16 × 16, and approximately 40% of the patches are randomly selected for masking. These masked positions are denoted as $M$.

To handle the masked patches, a shared learnable embedding $e_{[M]}$ is used to replace the original image patch embeddings $e_i^p$ if $i \in M$, as shown in the following Eq. 1:

$$x_i^M = \delta(i \in M) \odot e_{[M]} + (1 - \delta(i \in M) \odot x_i^p \qquad (1)$$

where $\delta(\cdot)$ is the indicator function and $\odot$ demotes element-wise multiplication.

Subsequently, a learnable CLS token is prepended to the input, which now becomes $[e_{CLS}, \{x_i^M\}_{i=1}^N]$. This sequence is then fed to the ViT[4] block, where the final encoding vectors are denoted as $\{h_i\}_{i=0}^N$, with $h_0$ corresponding to the CLS token. The visual tokens of the masked positions are

then predicted based on the corrupted image $x^M$, and a simple fully connected layer is used for this prediction.

When using the BEiT-V2 model as the image encoder layer, a distinction from the pre-training stage lies in the handling of image patches. Specifically, after the input image $x$ is segmented into patches, these patches are not masked. Instead, the image is processed directly by prepending a learnable CLS token to the input, as shown in Eq. 2. This CLS token, denoted as $e_{CLS}$.

$$z_0 = \left[ e_{CLS}, \ \{ x_i^p \}_{i=1}^N \right] \quad (2)$$

where $x^p \in R^{N \times (P^2 C)}$.

The image feature vector $y = \{H_i\}_{i=0}^N$ (where $H_0$ corresponds to the CLS token) is derived through the ViT block (Eqs. 3–5), which is equivalent to the Transformer encoder layer. This layer alternates between multi-headed self-attention (MSA) and Multi-Layer Perceptron (MLP) blocks. Specifically, Eq. 3 describes the MSA operation, Eq. 4 the MLP operation, and Eq. 5 provides the final output after the last layer:

$$z_l' = MSA\left(LN\left(z_{l-1}\right)\right) + z_{l-1}, \ l = 1, \dots, L \quad (3)$$

$$z_l = MLP\left(LN\left(z_l'\right)\right) + z_l', \ l = 1, \dots, L \quad (4)$$

$$y = ViT_L(x) = LN\left(z_L\right) \quad (5)$$

The MSA block extends self-attention (SA, Eqs. 6 and 7) by running $k$ self-attention operations (or "heads") and concatenating their outputs (Eq. 8). Layer Norm is applied before each block, with residual connections following each block.

$$\left[ \boldsymbol{q}, \boldsymbol{k}, \boldsymbol{v} \right] = \boldsymbol{x} U_{qkv}, \ U_{qkv} \in R^{D \times 3D_h}, \ x \in R^{N \times D} \quad (6)$$

$$SA(\boldsymbol{x}) = softmax\left( \frac{\boldsymbol{q}\boldsymbol{k}^T}{\sqrt{D_h}} \right) \boldsymbol{v} \quad (7)$$

$$MSA(\boldsymbol{x}) = \left[ SA_1(x); SA_2(x); \dots; SA_k(x) \right] U_{msa}, \ U_{msa} \in R^{k \cdot D_h \times D} \quad (8)$$

(2) Spatial Encoder Layer: we acquire the image CLS token $H_0$, a representation of the global image, after the original image has passed through the image encoder layer. We employ a two-layer ViT block (Eq. 5, $L = 2$) as the spatial encoding layer to further extract spatial information from the sample. Specifically, we use the multi-dimensional vector of the image CLS token as input, obtained by encoding the same sample at different spatial heights. We then prepend a new learnable CLS token to this input before feeding it into the spatial encoder layer. This process yields the spatial image CLS token enriched with spatial image features.

(3) Temporal Encoder Layer: we utilize a Bidirectional Long Short-Term Memory (Bi-LSTM) layer, which processes the input sequence in both forward and backward directions simultaneously. This bidirectional approach allows the Bi-LSTM to capture information from both past (previous time steps) and future (following time steps), effectively capturing long-term dependencies in the input sequence. After obtaining the multi-dimensional vector of CLS token from the spatial encoding layer across different time points of the same sample, we derive a vector containing both temporal and spatial features, suitable for the subsequent classification task.

(4) Classification Layer: we employ a simple MLP to implement three specific classifications, i.e., vMost-, vLess-, and vNo-DILI concern.

We use Cross Entropy as the loss function and Adaptive Moment Estimation with decoupled weight decay (AdamW) as the optimizer to guide model training, setting $lr = 2e - 4$, $batch\ size = 6$, $epoch = 100$ for training. To eliminate the model's dependence on the test dataset and ensure the reliability of the model performance indicators, we employ $K$-fold cross-validation ($K = 5$) during training, with the training dataset comprising 80% of the data through random partitioning. This approach allows all data to be used as test datasets for validation, and the average result served as the model evaluation indicator. The entire DILI classification prediction model is implemented using the PyTorch framework with Cuda 11.6 and trained on a Tesla V100 GPU (32GB).

### Reporting summary
Further information on research design is available in the Nature Portfolio Reporting Summary linked to this article.

### Data availability
The source data can be obtained from Supplementary Data 1–3. Supplementary Data 4 contains the CSV index file for the organoid image dataset, including metadata and classification labels for all images used in the study. The corresponding compressed image archive has not been deposited in a public repository, but the full dataset is available from the corresponding author upon reasonable request.

### Code availability
The code involved in this article is available on GitHub (https://github.com/js-ish/dilipredict).

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

## Acknowledgements

This research was funded by the Natural Science Foundation of Jiangsu Province, Major Project (BK20222008), Jiangsu Province Hospital (the First Affiliated Hospital with Nanjing Medical University) Clinical Capacity Enhancement Project (JSPH-MB-2023-9), and Open Project of Key Laboratory of Environmental Medicine Engineering of Ministry of Education (2024EME003).

## Author contributions

J.Z. and Z.G. designed the study and supervised the work. S.T., T.X., and W.W. performed organoid culture, toxicity detection, and experimental data analysis. J.R. and F.C. participated in the liver organoid methods and provided clinical insights. X.Y. and X.W. selected a test list of drugs. Y.D., Q.Z., T.H., Q.H., and Z.Y. established the AI model. M.L., P.X., and Z.C. reviewed and commented on the paper from theoretical viewpoints. G.L. and Y.P. provided intellectual input in project development. S.T., Y.D., and Q.Z. wrote the manuscript that was reviewed and edited by all authors.

## Competing interests

The authors declare no competing interests.

## Inclusion and ethics

This study obtained informed consent and received ethical approval from the Ethics Committee of the First Affiliated Hospital of Nanjing Medical University (2021-SR-575).
