## [Transparent Peer Review file · Communications Biology]

Development of an AI Model for DILI-Level Prediction Using Liver Organoid Brightfield Images

Corresponding Author: Professor Juan Zhang

Version 0:

Reviewer comments:

Reviewer #2

(Remarks to the Author)

The authors present a tool for predicting drug-induced liver injury utilizing AI image analysis performed on liver organoid brightfield images. The authors describe this tool (DILItracer) as promising for predicting liver injury of drugs in development in their pre-clinical stage, utilizing less invasive methods such as BF imaging and AI computer vision. After establishing their model, the results of this analysis provide a ternary classification of hepatotoxicity based on the FDA DILI labels.

The authors present a clear objective of their work and their results have good interpretability in regards to its potential clinical application.

This work also adds another application for utilizing organoids as in-vitro models for drug development and clinical correlation proving the various advantages of organoids due to their characteristics of maintaining fidelity to their source and adaptability for high-throughput testing.

The application of AI analysis on brightfield images of various tissue organoids is not novel and has been reported in other literature to quantify the growth of organoids. The authors do present this point and argue that drug-induced liver injury has not been of major focus. Therefore this work does present a potentially promising benefit as an add-on to current methods of pre-clinical drug testing.

Some comments on the content of the manuscript that warrant further clarification or modification:

1) The results especially in Figure 2 (a, g), lacked clear images of the morphology of the HLO and spheroids in the various conditions of the drugs they were exposed to (No, Most, Some DILI). As the premise of the work is to identify morphological changes in the HLOs captured on BF images for AI model training and prediction, adding additional details (both in text and figures) on the specific morphology of organoids under the different tested DILI conditions will help further understand the scalability of the work.

2) Figure 2 felt crowded so perhaps a dedicated figure for the BF images (see comment 1) can separate the morphology of HLO and spheroids from the accuracy and AUC of the model.

3) Figure 2 (a, g) and Figure 3 (h, i) lacked a scale bar.

4) Small typo in Figure 2 (c, i) "overall" instead of "overall" in the first bar.

5) The section in which HLOs and HepG2 spheroids are compared with their cytochrome P450 and markers (lines 162-174) might be more appropriate in an earlier section of the results section in the manuscript as an introduction to how HLOs were superior to the spheroids in their relevance as an in-vitro liver model.

6) How consistent were the ATP and ALB assays and the DILI status of the tested drugs? For example in supplementary Fig 1 and 2, Vinblastine and Digoxin labeled as vNo-DILI-Concern showed significant ALB and ATP inhibition, whereas Isoniazid labeled as vMost-DILI_Concern did not show a significant difference. Did HLOs treated with those medications follow the same trend as these assays in terms of their morphological appearance? And how does this discrepancy affect the training of the model?

Reviewer #3

(Remarks to the Author)

The study in consideration presents the development and evaluation of an AI-based drug-induced liver injury (DILI) risk prediction system using human liver organoids (HLOs) and brightfield imaging. The study uses a novel approach in context of DILI prediction by leveraging HLOs, which are more physiologically relevant compared to traditional models (e.g., HepG2 spheroids). The integration of multicellular clusters with liver-specific markers and metabolic enzymes (e.g., CYP enzymes) adds significant originality. Using a ternary classification (No-, Less-, Most-DILI concern) instead of the traditional binary outputs (e.g., toxic or non-toxic), increases granularity and clinical applicability.

The use of a Video Vision Transformer (ViViT)-inspired architecture to capture spatial and temporal features is an innovative application of deep learning to 3D organoid imaging. The combination of spatial and temporal encoding for DILI prediction is relatively unexplored in existing literature. Other features like Visualizing attention maps to understand the morphological changes contributing to DILI predictions is a unique addition, offering interpretability that many AI models lack. By training the model on labels derived from the FDA DILI rank database, the study ensures clinical relevance, which is a distinguishing factor compared to other studies that often use chemically or biologically derived labels.

Section wise comments:

INTRODUCTION

1. Reduce Repetition: Some points are repeated, such as the advantages of HLOs and brightfield imaging. Streamlining these would improve the flow. For example, mention the advantages of brightfield imaging only once in the section discussing image analysis.
2. Simplify dense sentences like "As preclinical identification of compounds at DILI risk remains a significant challenge in drug discovery, it is imperative to explore alternative in vitro strategies..."
3. Highlight the performance of DILITracer earlier to capture the reader's attention. For instance: "Our model, DILITracer, achieved a remarkable 82.34% accuracy, particularly excelling in identifying non-DILI compounds (90.16%)."

METHODOLOGY

1. Organize subsections clearly. Divide the methodology with subheadings, such as: HLO and HepG2 Spheroid Culture, Compound Screening, Data Collection and Reprocessing, Model Architecture, Training, and Validation.
2. Explain in brief the choice of tools and parameters, like why a 224×224 resolution or a patch size of 16×16 was selected. Justify the choice of BEiT-V2 over other image encoders and discuss its specific benefits in this context.
3. Clarify Drug Testing and Concentrations: The explanation of 0, 1, 10, and 100×C_{max} could benefit from a brief rationale for why these concentration gradients were chosen. Highlight why 10×C_{max} was identified as the optimal concentration for the DILI assay on HLOs.
4. Mathematical formulations may overwhelm some readers. Add intuitive explanations alongside the equations.
5. Mention why the chosen embedding and culture methods (e.g., Matrigel, ultra-low attachment microplates) are suitable for the experiments.
6. Highlight the significance of the DILIRank classification system and how it ensures the biological relevance of the chosen compounds.
7. Discuss limitations or challenges, if any, in the methods used, briefly, like challenges faced in the experimental or computational setup.

RESULTS

The results are detailed, with clear comparisons between the two experimental platforms (HLOs and HepG2 spheroids).

1. Highlight the 90.16% accuracy for vNo-DILI cases earlier to draw attention to the model's strong points.
2. Present the metrics (accuracy, recall, specificity, F1 score) for all categories in a table for better visualization and comparison.

DISCUSSION

The discussion incorporates quantitative metrics like accuracy, recall, specificity, and F1 scores, providing a robust evaluation of the model's performance.

1. Start with a summary of the key findings (e.g., the superior performance of HLOs, the predictive accuracy, and clinical relevance). Follow with a comparison to previous studies, highlighting advancements and differences. Expand clinical relevance.
2. Elaborate on how this model could influence preclinical testing workflows and reduce the incidence of clinical trial failures due to DILI. You can discuss potential integration with existing drug safety pipelines.
3. Presentation of Data: Include more visual aids, such as tables or concise charts, to summarize key results (e.g., comparison of HLO and HepG2 spheroid performance, ablation study results, or enzyme expression levels).

CONCLUSION

1. Summarize the overall contribution of the study in a concise paragraph, emphasizing the novelty of the approach, its practical relevance, and the potential for future refinement.

Version 1:

Reviewer comments:

Reviewer #2

(Remarks to the Author)

The authors present a tool for predicting drug-induced liver injury utilizing AI image analysis performed on liver organoid brightfield images. The authors describe this tool (DILTracer) as promising for predicting liver injury of drugs in development in their pre-clinical stage, utilizing less invasive methods such as BF imaging and AI computer vision. After establishing their model, the results of this analysis provide a ternary classification of hepatotoxicity based on the FDA DILI labels. The authors present a clear objective of their work and their results have good interpretability in regards to its potential clinical application.

This work also adds another application for utilizing organoids as in-vitro models for drug development and clinical correlation proving the various advantages of organoids due to their characteristics of maintaining fidelity to their source and adaptability for high-throughput testing.

The application of AI analysis on brightfield images of various tissue organoids is not novel and has been reported in other literature to quantify the growth of organoids. The authors do present this point and argue that drug-induced liver injury has not been of major focus. Therefore, this work does present a potentially promising benefit as an add-on to current methods of pre-clinical drug testing.

In this revised manuscript, the authors addressed my comments from the first review. Figure 2 includes larger photos showcasing the HLO's under brightfield imaging under different treatments.

The re-organization of the text makes it smoother to read and follow the sequence of the experiments and results. The discussion section highlights some important limitations of the study but also in my opinion promising applications of this tool such as the ability to identify critical time points of the drug effects on the organoids given the non-invasive nature of BF imaging.

I have no further comments or reviews.

Reviewer #3

(Remarks to the Author)

Thank you for your hard work on this manuscript. I have reviewed the latest version, and all suggested changes have been successfully incorporated. The manuscript is now in great shape and ready for publication.

Reviewer #2 (Remarks to the Author):

The authors present a tool for predicting drug-induced liver injury utilizing AI image analysis performed on liver organoid brightfield images. The authors describe this tool (DILTracer) as promising for predicting liver injury of drugs in development in their pre-clinical stage, utilizing less invasive methods such as BF imaging and AI computer vision. After establishing their model, the results of this analysis provide a ternary classification of hepatotoxicity based on the FDA DILI labels.

The authors present a clear objective of their work and their results have good interpretability in regards to its potential clinical application.

This work also adds another application for utilizing organoids as in-vitro models for drug development and clinical correlation proving the various advantages of organoids due to their characteristics of maintaining fidelity to their source and adaptability for high-throughput testing.

The application of AI analysis on brightfield images of various tissue organoids is not novel and has been reported in other literature to quantify the growth of organoids. The authors do present this point and argue that drug-induced liver injury has not been of major focus. Therefore this work does present a potentially promising benefit as an add-on to current methods of pre-clinical drug testing.

Some comments on the content of the manuscript that warrant further clarification or modification:

1) The results especially in Figure 2 (a, g), lacked clear images of the morphology of the HLO and spheroids in the various conditions of the drugs they were exposed to (No, Most, Some DILI). As the premise of the work is to identify morphological changes in the HLOs captured on BF images for AI model training and prediction, adding additional details (both in text and figures) on the specific morphology of organoids under the different tested DILI conditions will help further understand the scalability of the work.

■ Response:

Thank you for your valuable suggestion. To enhance clarity, we reduced the number of BF images from 3*4 to 2*2 (per day) across different z-axes (with the remaining images included in FigS5 and FigS6).

In response to comment 2, we have separated BF images and platform stability markers (such as ALB and ATP) from the predictive performance results of the model into two distinct figures. To emphasize morphological details of organoids under different DILI conditions, we have added stacked images along with corresponding description.

■ Revised manuscript (Page 5, Line 122-130):

... Taking the HLO-based DILI toxicity platform as an example, the significant difference by compounds classified at different levels of liver toxicity potency could be observed. When treated with chlorpheniramine, labeled “No-DILI” by DILRank, HLOs still increased in diameter and developed into a typical translucent hollow sphere with clear boundaries. In contrast to non-hepatotoxic compounds, Gefitinib-stimulated HLOs, labeled as “Most-DILI”, underwent cell death, failed to maintain their

original spherical structure, and disintegrated by the end of Day 3. The state of HLOs treated with Simvastatin (with a label of “Less-DILI”) was between No- and Most-DILI, i.e. HLOs showed growth inhibition but their morphology was still in the form of a complete sphere. Overall, we provided a robust biological basis for the subsequent development of DILI risk prediction models (Fig. 2a, c, e).

2) Figure 2 felt crowded so perhaps a dedicated figure for the BF images (see comment 1) can separate the morphology of HLO and spheroids from the accuracy and AUC of the model.

■ **Response:**

In response to comments 1 and 2, we have revised the corresponding figures (please see Figure 2 and Figure 3) to address the feedback.

Fig2. DILI toxicity testing platform based on human liver organoids models. a, c, e, brightfield

morphology changes (12 original images and 1 stacked image) of three compounds at different levels of liver injury in human liver organoids (Scale bar = 100 μm) from Day 0 to Day 3. **b, d, f**, ALB inhibition and ATP inhibition of three compounds at different levels of liver injury in human liver organoids at end of Day 3. ** $P < 0.01$; **** $P < 0.0001$. $n = 3$.

Fig3. DILI toxicity testing platform based on HepG2 spheroids models. **a, c, e**, brightfield morphology changes (12 original images and 1 stacked image) of three compounds at different levels of liver injury in HepG2 spheroids (Scale bar = 250 μm) from Day 0 to Day 3. **b, d, f**, ALB inhibition and ATP inhibition of three compounds at different levels of liver injury in HepG2 spheroids at end of Day 3. ** $P < 0.01$; **** $P < 0.0001$. $n = 3$.

3) Figure 2 (a, g) and Figure 3 (h, i) lacked a scale bar.

■ **Response:**

Thank you for pointing this out. The scale bar was already included in the figures, but due to the image clarity, it may have been difficult to notice. We have now adjusted the image resolution to improve clarity, making the scale bar more visible.

4) Small typo in Figure 2 (c, i) “overall” instead of “overall” in the first bar.

■ **Response:**

Apologies for the typo in Figure 2 (c, i) and thank you for pointing it out. We have corrected the misspelling of “ovreal” to “overall” in both bar charts.

5) The section in which HLOs and HepG2 spheroids are compared with their cytochrome P450 and markers (lines 162-174) might be more appropriate in an earlier section of the results section in the manuscript as an introduction to how HLOs were superior to the spheroids in their relevance as an in-vitro liver model.

■ **Response:**

Thank you for your valuable suggestion. We have moved the comparison of cytochrome P450 between HLOs and HepG2 spheroids to an earlier section of the manuscript. However, we intentionally placed this comparison after the discussion of the model prediction accuracy. Once we demonstrated the predictive capabilities of the model, we then explored the biological advantages of HLOs in terms of cytochrome P450 expression. This logical flow allowed us to introduce the biological aspects of the HLO model as a supportive factor that enhances the overall prediction accuracy. Additionally, the ablation experiments were conducted after establishing the biological advantages to further validate the importance of temporal and spatial modalities in the prediction model. We believe this structure provides a more cohesive narrative, where the biological advantages of HLOs enhance the model’s effectiveness and predictive accuracy. Thank you once again for your insightful feedback.

■ **Revised manuscript (Page 9-10, Line 179-205):**

Superiority analysis of the DILI prediction model using human liver organoids from *in vitro* and *in silico* perspectives

Next, we attempted to demonstrate the superiority of our AI model from the perspective of *in vitro* biological models. The result of immunofluorescence (Fig. 5a) confirmed liver-specific ‘bile duct-like structure’ as indicated by the markers of bile salt export pump (BSEP). Also, tight junction protein stained by zonula occludens-1 (ZO-1) suggested a multi-cellular type 3D hollow body, including hepatocytes rich with hepatocyte nuclear factor 4-alpha (HNF4a), CD31 expressing Liver Sinusoidal Endothelial Cells (LSECs), CD68⁺ Kupffer cells (KCs), and DES-containing Hepatic Stellate Cells (HSCs). Importantly, the HLO model showed significantly higher expression levels of metabolic enzyme *CYP3A4* (1.80 folds), *CYP1A2* (88.96 folds), *CYP2D6* (4.95 folds), *CYP2E1* (10.79 folds), *CYP2C9* (8.16 folds), and *CYP2C19* (8.62 folds) than those in HepG2 spheroids (Fig. 5b). Therefore, we assumed that liver organoids, as a more physiologically relevant *in vitro* liver model, would be able to generate more realistic toxicological responses and thus provide more reliable image data for the development of

DILI models.

We further conducted ablation experiments to investigate the impact of temporal and spatial modalities on the DILI prediction model (Table. 2). First, we used Day 0-Day 1 image data instead of Day 0-Day 3 images to evaluate the role of temporal dimension information. The results showed a 12.09% decrease in prediction accuracy compared to the original model, with lower recall, specificity, precision, and F1 scores for each label (Fig. 5c). Second, we replaced multiple separate images of the same sample taken at different heights with 3D composite images generated by a fusion algorithm (provided by the HCS instrument), which combines images from different focal planes. After removing the spatial coding layer from the model, the prediction accuracy dropped by 6.24% compared to the original model. Recall, precision, and F1 scores for each label did not surpass the original model's metrics (Fig. 5d). Overall, both temporal and spatial modalities exerted a substantial positive influence on the development of DILI prediction models, thus not only validating the soundness and effectiveness of the modeling approach but also emphasizing the pivotal role of temporal and spatial dimensions in replicating intricate biological mechanisms.

Fig5. Superiority analysis and ability validation of human liver organoids image-only model. a, immunofluorescence detection of ZO-1, BSEP, HNF4 α , DES, CD31, CD68 in human liver organoids. Scale bar = 50 μ m. **b,** expression level of *CYP3A4*, *CYP1A2*, *CYP2D6*, *CYP2E1*, *CYP2C9*, and *CYP2C19* between HepG2 spheroid and HLOs. * $P < 0.05$, ** $P < 0.01$; **** $P < 0.0001$. $n = 3$. **c,** radar

plot visualizing the confusion matrix in ablation experiments based on spatial dimensions and Receiver Operating Characteristic (ROC) curves of human liver organoids image only model using Day0-Day1 images. **d**, radar plot visualizing the confusion matrix in ablation experiments based on temporal dimensions and ROC curves of human liver organoids image only model without the structure of the spatial coding layer. Green, yellow, and blue curves represent vNo-, vLess-, and vMost-DILI-Concern labels, respectively. **e**, attention heatmap of Gefitinib. z-axis (top image) = 135,650; z-axis (bottom image) = 135,950; **f**, attention heatmap of Chlorphenamine. z-axis (top image) = 136,500; z-axis (bottom image) = 136,900; The red and the blue areas in the heatmap indicate the areas with higher and lower attention weights respectively. The small boxes mark the areas that are in focus precisely under each visual field. The large red boxes mark the time period with higher activation of attention weights. Scale bar = 100 μm .

6) How consistent were the ATP and ALB assays and the DILI status of the tested drugs? For example in supplementary Fig 1 and 2, Vinblastine and Digoxin labeled as vNo-DILI-Concern showed significant ALB and ATP inhibition, whereas Isoniazid labeled as vMost-DILI_Concern did not show a significant difference. Did HLOs treated with those medications follow the same trend as these assays in terms of their morphological appearance? And how does this discrepancy affect the training of the model?

■ **Response:**

Thank you for your detailed and valuable feedback. We would like to elaborate on the rationale for selecting ATP and ALB assays, as well as how they relate to the brightfield imaging data:

Selection of ATP and ALB as Auxiliary Methods Without Placing Excessive Emphasis on Their Consistency with Brightfield Imaging: ALB production involves hepatocellular transcription of prepro-albumin, translation, N-terminal cleavage to form pro-albumin, release from the rough endoplasmic reticulum, and final cleavage in the Golgi apparatus before secretion into serum^[1]. Additionally, ATP detection using the CTG assay is widely applied to assess mitochondrial activity and function. Together, these two biomarkers reflect a broad range of hepatocellular functionality and provide insights into hepatocyte health. This informed our choice of ALB and ATP as auxiliary methods to confirm that the organoids could exhibit measurable biochemical changes. **These two markers were not intended to directly correlate with brightfield imaging data but to serve as supplementary evidence supporting the reliability of the organoid model for DILI prediction. The focus was not on the direct consistency between brightfield images and biochemical markers, but on demonstrating biochemical changes in the organoids, which would serve as the basis for subsequent image analysis.**

Differences in Mechanistic Reflection and Limitations of Single Concentration: The lack of significant changes in ATP and ALB in organoids treated with Isoniazid may reflect that Isoniazid, as a typical drug of idiosyncratic DILI, may primarily engage other signaling pathways rather than those linked to mitochondrial activity and synthetic function at the tested concentrations. In contrast, brightfield imaging offers a broader view of organoid health. By incorporating spatial and temporal

features into a DILI prediction model, brightfield imaging facilitates a more accurate assessment of DILI, providing a comprehensive and dynamic understanding of the drug's effects within the organoid model. Moreover, while the prediction rate at the 10*C_{max} concentration was relatively high, we acknowledge that Digoxin and Vinblastine may have been tested at concentrations that could potentially trigger toxicity responses. However, 10*C_{max} was identified as the most optimal concentration for maximizing prediction accuracy. These compounds were not excluded from the model to ensure sufficient data volume and learning capacity, despite the potential for high-concentration toxicity in this study.

Future Directions for Incorporating More Mechanistic Feedback: In this study, due to considerations of practicality applicability, convenience, and predictive performance, we selected the continuously captured brightfield images of organoids under different DILI condition as data modality and aimed to excavate the spatial and temporal features of brightfield images. We acknowledged that ATP and ALB alone are insufficient to capture the complete range of biochemical responses, and in the future, we plan to expand our approach to include a broader range of biochemical markers that can reflect more comprehensive DILI mechanisms such as mitochondrial injury, reactive metabolites, biliary transport inhibition, and immune responses. This model will provide deeper mechanistic insights into liver injury, ultimately contributing to safer drug development and more informed clinical decision-making.

Ref:

[1] Baudy, A. R., Otieno, M. A., Hewitt, P., Gan, J., Roth, A., Keller, D., Sura, R., Van Vleet, T. R., and Proctor, W. R. (2020). Liver Microphysiological Systems Development Guidelines for Safety Risk Assessment in the Pharmaceutical Industry. *Lab Chip*, 20(2), 215–225. <https://doi.org/10.1039/c9lc00768g>

Reviewer #3 (Remarks to the Author):

The study in consideration presents the development and evaluation of an AI-based drug-induced liver injury (DILI) risk prediction system using human liver organoids (HLOs) and brightfield imaging. The study uses a novel approach in context of DILI prediction by leveraging HLOs, which are more physiologically relevant compared to traditional models (e.g., HepG2 spheroids). The integration of multicellular clusters with liver-specific markers and metabolic enzymes (e.g., CYP enzymes) adds significant originality. Using a ternary classification (No-, Less-, Most-DILI concern) instead of the traditional binary outputs (e.g., toxic or non-toxic), increases granularity and clinical applicability.

The use of a Video Vision Transformer (ViViT)-inspired architecture to capture spatial and temporal features is an innovative application of deep learning to 3D organoid imaging. The combination of spatial and temporal encoding for DILI prediction is relatively unexplored in existing literature. Other features like Visualizing attention maps to understand the morphological changes contributing to DILI predictions is a unique addition, offering interpretability that many

AI models lack. By training the model on labels derived from the FDA DILI rank database, the study ensures clinical relevance, which is a distinguishing factor compared to other studies that often use chemically or biologically derived labels.

Section wise comments:

INTRODUCTION

1. Reduce Repetition: Some points are repeated, such as the advantages of HLOs and brightfield imaging. Streamlining these would improve the flow. For example, mention the advantages of brightfield imaging only once in the section discussing image analysis.

■ **Response:**

Thank you for your valuable suggestion. We have streamlined the manuscript to reduce repetition, particularly concerning the advantages of HLOs and brightfield imaging.

2. Simplify dense sentences like “As preclinical identification of compounds at DILI risk remains a significant challenge in drug discovery, it is imperative to explore alternative *in vitro* strategies...”

■ **Response:**

We appreciate your feedback. We have simplified the sentence structure to improve readability. In response to the comment 1 and 2, we have revised our manuscript and marked them in yellow. The revised manuscript as followed:

■ **Revised manuscript (Page 1-2, Line 34-73):**

The computer vision (CV) model has shown significant potential in clinical applications by enabling detailed analysis of complex visual information from cell images¹. Recently, vision transformer (ViT) has made breakthrough progress in the field of CV, signaling a transition from the Convolutional Neural Network to the Transformer backbone^{2,3}. ViT utilizes a self-attention mechanism to capture long-distance relationships in an image, enabling it to understand global dependencies in the data⁴. This ability to capture the overall structure of biomedical images is why we selected ViT for this work. CV models using 2D biomedical images have delivered impressive predictive capabilities in tasks such as detecting cell death⁵, segmenting cell nuclei⁶, and localizing subcellular protein⁷—achievements that are difficult with manual analysis. However, the emergence of physiologically relevant 3D models like spheroids^{8,9} and organoids¹⁰, underscores the urgent need for the development of advanced 3D imaging techniques and novel cell morphology analysis algorithms in CV. Currently, while drug screening based on phenotypes or statuses from cell images has gradually been applied, there has been less focus on identifying drug-induced liver injury (DILI). This may be due to the substantial metabolic differences between humans and animals¹¹, making it difficult to reflect the actual effects of compounds. For example, traditional preclinical safety trials of 150 drugs reported predictive accuracies of only 63% and 43% in non-rodent and rodent animals, respectively, with the lowest accuracy observed in the hepatobiliary system¹². Preclinical identification of DILI-risk compounds remains a challenge in drug discovery^{13,14}, emphasizing the need for alternative *in vitro* strategies to assess hepatotoxicity compounds and generate reliable data for CV model development. The aim of this work is to develop an

expeditious tool based on CV model for preclinical even clinical drug safety assessment.

Organoid culture technology provided new experimentally tractable, physiologically relevant models of human pathologies and subsequent drug screening¹⁵. Human liver organoids (HLOs) offer distinct advantages over HepG2 spheroids, as they comprise both hepatic parenchymal and non-parenchymal cells, reflecting accurate intercellular interactions. As 3D multicellular clusters, HLOs carry a cytochrome P450 system involved in drug metabolism, and preserve phenotype and function of hepatocyte longer than primary human hepatocytes (PHHs)^{16,17}. Recent studies highlight the potential of AI-driven image processing to explore the strong correlation between organoid morphology and compound toxicity or disease status^{18,19}. Therefore, the organoid model is not only a viable alternative to animal model but also a promising tool primed for assessing DILI risks through morphological analysis. In image analysis, dyes are frequently used to highlight cell features and CV techniques are then utilized to identify any changes²⁰. Notably, brightfield imaging surpasses fluorescence imaging in several aspects: real-time capabilities, non-destructive nature, and the absence of additional sample processing requirements. Furthermore, brightfield imaging excels in information retrieval due to its high capacity, richness, and depth, all while being cost- and time-effective. To capture the 3D features of the organoid model, we applied 3D video processing principles and developed a CV model based on image-spatial-temporal coding layers to extract spatiotemporal information from high-content screening (HCS). Herein, we developed an evaluation system, named DILITracer, capable of predicting the clinical DILI of compounds based on HLO technology platform and AI-assisted algorithm for data analysis. The model achieved an impressive overall accuracy of 82.34%, with particularly high accuracy (90.16%) in identifying non-DILI compounds.

3. Highlight the performance of DILITracer earlier to capture the reader's attention. For instance: "Our model, DILITracer, achieved a remarkable 82.34% accuracy, particularly excelling in identifying non-DILI compounds (90.16%)."

■ **Response:**

Thank you. In our revision, we placed the performance results of DILITracer at the end of the second paragraph in Introduction section, highlighting the impressive accuracy of the model. We begin with the background and motivation before presenting the model's performance, believing this structure effectively guides the reader's attention to the model's capabilities.

■ **Revised manuscript (Page 2, Line 69-73):**

... Herein, we developed an evaluation system, named DILITracer, capable of predicting the clinical DILI of compounds based on HLO technology platform and AI-assisted algorithm for data analysis. The model achieved an impressive overall accuracy of 82.34%, with particularly high accuracy (90.16%) in identifying non-DILI compounds.

To our knowledge, DILITracer is the 1st model able to categorize hepatotoxicity levels (no, less, or most DILI levels) rather than merely dictating hepatotoxicity. ...

METHODOLOGY

1. Organize subsections clearly. Divide the methodology with subheadings, such as: HLO and

HepG2 Spheroid Culture, Compound Screening, Data Collection and Reprocessing, Model Architecture, Training, and Validation.

■ **Response:**

Thank you for your thoughtful suggestion. We have reorganized the subheadings in the methodology accordingly, as follows: ‘Culture of Human Liver Organoids and HepG2 Spheroids’; ‘Compounds Screening’; ‘DILI Toxicity Test’; ‘Data Collecting, Labelling, and Pre-processing’; ‘Model Architecture, Training, and Validation’. We have also marked them in yellow in the revised paper.

2. Explain in brief the choice of tools and parameters, like why a 224×224 resolution or a patch size of 16×16 was selected. Justify the choice of BEiT-V2 over other image encoders and discuss its specific benefits in this context.

■ **Response:**

Thank you for your thoughtful suggestion.

A **224×224 image resolution** has become a common standard in deep learning, offering an ideal balance between computational efficiency and the retention of image detail. This resolution is particularly effective in capturing key features, such as the subtle morphological variations of liver organoids in this study. It ensures that essential details are preserved while keeping computational demands manageable. In summary, the 224×224 resolution is widely adopted across both 2D and 3D imaging applications, providing sufficient detail without overwhelming computational resources [1-3].

A **patch size of 16×16** is optimal for vision transformers like BEiT-V2, which divide images into smaller sections for parallel processing. This smaller patch size enables the model to focus on localized features, which is essential for detecting fine-grained differences in cellular structures and damage [4], such as hepatotoxicity. This approach allows the model to capture fine-grained details while maintaining computational efficiency. Therefore, the choice of a 16×16 patch size is well-justified and aligns with the design principles of BEiT-V2.

Specific benefits of BEiT-V2 over other image encoders lie in:

(1) Pretraining on Large-Scale Data: In this study, BEiT-V2 has been pretrained on a massive dataset of 700,000 cell images, showcasing its strong ability to handle large-scale data effectively [4]. This pretraining enables BEiT-V2 to learn robust, generalized features across diverse biological contexts, facilitating effective knowledge transfer to new datasets, such as liver organoid images.

(2) 3D Feature Extraction: Thanks to BEiT-V2’s ability to transform images into semantically rich visual tokens, it is particularly well-suited for applications involving 3D images [5]. This capability is essential for understanding the multi-dimensional nature of organoid images, as it enables the capture of morphological changes from deep-dimensional features under different DILI conditions.

Ref:

- [1] Tayebi Arasteh, S., Misera, L., Kather, J.N., et al. (2024). Enhancing diagnostic deep learning via self-supervised pretraining on large-scale, unlabeled non-medical images. *Eur Radiol Exp*, 8, 10. <https://doi.org/10.1186/s41747-023-00411-3>
- [2] Talebi, H., and Milanfar, P. (2021). Learning to Resize Images for Computer Vision Tasks. In *2021 IEEE/CVF International Conference on Computer Vision (ICCV)*, Montreal, QC, Canada (pp. 487-496). <https://doi.org/10.1109/ICCV48922.2021.00055>
- [3] Yavuz, M. C., and Yang, Y. (2025). Cross-D Conv: Cross-dimensional transferable knowledge base via Fourier shifting operation. *arXiv:2411.02441v5 [cs.CV]*. <https://arxiv.org/abs/2411.02441v5>
- [4] Peng, Z., Dong, L., Bao, H., Ye, Q., and Wei, F. (2022). BEiT v2: Masked Image Modeling with Vector-Quantized Visual Tokenizers. *arXiv:2208.06366v1 [cs.CV]*. <https://arxiv.org/pdf/2208.06366v1>
- [5] Peng, Z., Dong, L., Bao, H., Ye, Q., & Wei, F. (2021). BEIT: BERT Pre-Training of Image Transformers. *arXiv:2106.08254v1 [cs.CV]*. <https://arxiv.org/pdf/2106.08254v1>

3. Clarify Drug Testing and Concentrations: The explanation of 0, 1, 10, and 100×C_{max} could benefit from a brief rationale for why these concentration gradients were chosen. Highlight why 10×C_{max} was identified as the optimal concentration for the DILI assay on HLOs.

■ **Response:**

Thank you for your constructive suggestion. C_{max} can intuitively reflect the actual presence level of a drug in the body's circulatory system. In our *in vitro* experiments based on HepG2 spheroids, we used 1×C_{max} as a reference for the drug dosage to better approximate the *in vivo* conditions. On this basis, we selected 10×C_{max} as the reference concentration, taking into account the coefficient of variation in toxicology (individual variability: tenfold). Also, based on existing literature ^[1, 2], we included 100×C_{max} to account for the possibility of higher drug concentrations in extreme or therapeutic scenarios, thereby ensuring a more comprehensive assessment of drugs that may exhibit toxicity under elevated exposure conditions. In summary, the concentration gradients of 0, 1, 10, and 100×C_{max} were selected to cover a range of drug exposure scenarios.

DILI toxicity testing based on HepG2 spheroids was conducted using concentration gradients of 0, 1, 10, 100×C_{max} and then used the corresponding brightfield image data to train the ViLT and iRENE models. 10×C_{max} was identified as the optimal concentration based on results from both deep learning models (Tables S1 and S2). This concentration strikes a balance between accurately representing clinical drug exposures and effectively capturing toxicological effects *in vitro*. From a biological standpoint, 1×C_{max} was too low to induce sufficient cellular damage to reveal subtle toxic effects. 100×C_{max} represented an extreme concentration that, while useful for evaluating high-exposure toxicity scenarios, is too high to reflect typical *in vivo* conditions. In summary, we selected 10×C_{max} as an optimal concentration for the DILI assay on organoids to provide a more

realistic drug exposure level to some extent.

We have added corresponding content to the revision and marked them in yellow. Thank you again for your valuable input.

Table S1. HepG2 spheroid based on ViLT model with a classification of drug concentrations.

	1*Cmax	10*Cmax	100*Cmax
5-fold cross-validation(1) Training set (best/last)	1/1	1/1	1/1
5-fold cross-validation(1) Test set (best/last)	0.72/0.68	0.75/0.71	0.70/0.60
5-fold cross-validation(2) Training set (best/last)	1/1	1/1	1/1
5-fold cross-validation(2) Test set (best/last)	0.72/0.56	0.75/0.54	0.75/0.60
5-fold cross-validation(3) Training set (best/last)	1/1	1/1	1/1
5-fold cross-validation(3) Test set (best/last)	0.80/0.68	0.83/0.75	0.65/0.65
5-fold cross-validation(4) Training set (best/last)	1/1	1/1	1/1
5-fold cross-validation(4) Test set (best/last)	0.72/0.6	0.75/0.46	0.65/0.45
5-fold cross-validation(5) Training set (best/last)	1/1	1/1	1/1
5-fold cross-validation(5) Test set (best/last)	0.73/0.59	0.64/0.59	0.69/0.50
Training set Ave (best/last)	1/1	1/1	1/1
Test set Ave (best/last)	0.73/0.62	0.74/0.61	0.69/0.56

Table S2. HepG2 spheroid based on iRENE model with a classification of drug concentrations.

	1*Cmax	10*Cmax	100*Cmax
5-fold cross-validation(1) Training set (best/last)	1/1	1/1	1/1
5-fold cross-validation(1) Test set (best/last)	0.68/0.48	0.83/0.63	0.65/0.65
5-fold cross-validation(2) Training set (best/last)	1/1	1/1	1/1
5-fold cross-validation(2) Test set (best/last)	0.92/0.76	0.75/0.63	0.70/0.70
5-fold cross-validation(3) Training set (best/last)	1/1	1/1	1/1
5-fold cross-validation(3) Test set (best/last)	0.76/0.68	0.83/0.83	0.75/0.75
5-fold cross-validation(4) Training set (best/last)	1/1	1/1	1/1
5-fold cross-validation(4) Test set (best/last)	0.76/0.64	0.67/0.58	0.90/0.80
5-fold cross-validation(5) Training set (best/last)	1/1	1/1	1/1

5-fold cross-validation(5) Test set (best/last)	0.73/0.55	0.82/0.73	0.75/0.63
Training set Ave (best/last)	1/1	1/1	1/1
Test set Ave (best/last)	0.77/0.62	0.78/0.68	0.75/0.71

Ref:

- [1] Ewart, L., Apostolou, A., Briggs, S. A., et al. (2022). Performance assessment and economic analysis of a human liver-chip for predictive toxicology. *Communications Medicine*, 2(1), 154. <https://doi.org/10.1038/s43856-022-00209-1>
- [2] Xu, J. J., Henstock, P. V., Dunn, M. C., et al. (2008). Cellular imaging predictions of clinical drug-induced liver injury. *Toxicological sciences: an official journal of the Society of Toxicology*, 105(1), 97–105. <https://doi.org/10.1093/toxsci/kfn109>

■ **Revised manuscript (Page 16, Line 343-351):**

The peak serum concentration (C_{max}) can intuitively reflect the actual presence level of a drug in the body's circulatory system. Therefore, we used 1*C_{max} as a reference for the drug dosage to better approximate the *in vivo* conditions. On this basis, we selected 10*C_{max} as the reference concentration, taking into account the coefficient of variation in toxicology (individual variability: tenfold). Also, based on existing literature ^{26, 36}, we included 100*C_{max} to account for the possibility of higher drug concentrations in extreme or therapeutic scenarios, thereby ensuring a more comprehensive assessment of drugs that may exhibit toxicity under elevated exposure conditions. In summary, for HepG2 spheroids, the concentration gradients of 0, 1, 10, and 100*C_{max} were selected to cover a range of drug exposure scenarios. ...

4. Mathematical formulations may overwhelm some readers. Add intuitive explanations alongside the equations.

■ **Response:**

Thanks for your careful comment. We have made two key revisions: First, we have added more intuitive explanations in the text to clarify the concepts behind the mathematical formulations. Second, we have reorganized the structure by placing the relevant formulas immediately after the corresponding explanatory text, which improves the flow and clarity of the manuscript. The revised manuscript has marked in yellow.

■ **Revised manuscript (Page 17-18, Line 386-440):**

Model Architecture, Training, and Validation

The CV classification model employed in this study consisted of four layers: (1) Image Encoder Layer, (2) Spatial Encoder Layer, (3) Temporal Encoder Layer, and (4) the Classification Layer. The architecture is inspired by the Video Vision Transformer (ViViT) model ², which captures temporal dynamics in video data by treating it as sequences of images over time. Furthermore, we augment the image encoder layer with a spatial encoder layer to capture spatial relationships among images

positioned at different levels along the z-axis in three-dimensional space. This multi-layered approach strikes an optimal balance between model complexity and performance.

(1) Image Encoder Layer: the BEiT-V2³⁷ model is used to comprehensively capture image features. We initialize this layer with weights pretrained on a dataset of 700,000 cell images, leveraging the model’s ability to extract deep-dimensional features from cell images. The pre-training of the BEiT-V2 model occurs in two stages: the first stage involves Vector-quantized Knowledge Distillation (VQ-KD)³⁷, and the second stage trains the BEiT-V2 model itself using the visual symbols generated in the first stage. Visual symbols derived from VQ-KD serve as the training targets of the subsequent stage.

Contrastive language-image pre-training (CLIP)³⁸ is employed as the teacher model in the VQ-KD model of the first-stage. In the second-stage, for a given input image $x \in R^{H \times W \times C}$, it is reshaped to $N = HW/P^2$ patches $\{x_i^p\}_{i=1}^N$, where $x^p \in R^{N \times (P^2 C)}$ and (P, P) is the patch size. In this experiment, each 224×224 image is divided into a 14×14 grid of image patches, with each patch being 16×16 , and approximately 40% of the patches are randomly selected for masking. These masked positions are denoted as M .

To handle the masked patches, a shared learnable embedding $e_{[M]}$ is used to replace the original image patch embeddings e_i^p if $i \in M$, as shown in the following Eq. 1:

$$x_i^M = \delta(i \in M) \odot e_{[M]} + (1 - \delta(i \in M)) \odot x_i^p \quad (1)$$

where $\delta(\cdot)$ is the indicator function and \odot demotes element-wise multiplication.

Subsequently, a learnable CLS token is prepended to the input, which now becomes $[e_{CLS}, \{x_i^M\}_{i=1}^N]$. This sequence is then feed to the ViT⁴ block, where the final encoding vectors are denoted as $\{h_i\}_{i=0}^N$, with h_0 corresponding to the CLS token. The visual tokens of the masked positions are then predicted based on the corrupted image x^M , and a simple fully-connected layer is used for this prediction.

When using the BEiT-V2 model as the image encoder layer, a distinction from the pre-training stage lies in the handling of image patches. Specifically, after the input image x is segmented into patches, these patches are not masked. Instead, the image is processed directly by prepending a learnable CLS token to the input, as shown in Eq. 2. This CLS token, denoted as e_{CLS} .

$$z_0 = [e_{CLS}, \{x_i^p\}_{i=1}^N] \quad (2)$$

where $x^p \in R^{N \times (P^2 C)}$.

The image feature vector $y = \{H_i\}_{i=0}^N$ (where H_0 corresponds to the CLS token) is derived through the ViT block (Eq. 3-5), which is equivalent to the Transformer encoder layer. This layer alternates between multi-headed self-attention (MSA) and Multi-Layer Perceptron (MLP) blocks. Specifically, Eq. 3 describes the MSA operation, Eq. 4 the MLP operation, and Eq. 5 provides the final output after the last layer:

$$z'_l = MSA(LN(z_{l-1})) + z_{l-1}, \quad l = 1, \dots, L \quad (3)$$

$$z_l = MLP(LN(z'_l)) + z'_l, \quad l = 1, \dots, L \quad (4)$$

$$y = ViT_L(x) = LN(z_L) \quad (5)$$

The MSA block extends self-attention (SA, Eq. 6, 7) by running k self-attention operations (or “heads”) and concatenating their outputs (Eq. 8). Layer Norm is applied before each block, with residual

connections following each block.

$$[\mathbf{q}, \mathbf{k}, \mathbf{v}] = \mathbf{x}U_{qkv}, \quad U_{qkv} \in R^{D \times 3D_h}, \mathbf{x} \in R^{N \times D} \quad (6)$$

$$SA(\mathbf{x}) = \text{softmax}\left(\frac{\mathbf{q}\mathbf{k}^T}{\sqrt{D_h}}\right)\mathbf{v} \quad (7)$$

$$MSA(\mathbf{x}) = [SA_1(\mathbf{x}); SA_2(\mathbf{x}); \dots; SA_k(\mathbf{x})]U_{msa}, \quad U_{msa} \in R^{k \cdot D_h \times D} \quad (8)$$

(2) Spatial Encoder Layer: we acquire the image CLS token H_0 , a representation of the global image, after the original image passed through the image encoder layer. We employ a two-layer ViT block (Eq. 5, $L = 2$) as the spatial encoding layer to further extract spatial information from the sample. Specifically, we use the multi-dimensional vector of the image CLS token as input, obtained by encoding the same sample at different spatial heights. We then prepend a new learnable CLS token to this input before feeding it into the spatial encoder layer. This process yields the spatial image CLS token enriched with spatial image features.

5. Mention why the chosen embedding and culture methods (e.g., Matrigel, ultra-low attachment microplates) are suitable for the experiments.

■ **Response:**

Thank you for your detailed comment. I truly appreciate the opportunity to clarify the rationale behind the chosen embedding and culture methods.

The embedding method is currently the most widely used liver organoid culture method^[1], in this approach: the Matrigel/cell mixture is seeded in cell culture plates to enable the formation of dome-shaped structures. The function of Matrigel is to promote cell growth and the 3D characteristics of organoids^[2].

The low-adhesion U-bottom plate is specially treated to minimize cell adhesion and spreading, allowing gravity to guide cells to aggregate at the center of the U-shaped bottom, promoting tumor spheroid formation. This method has been successfully applied to 3D culture of HepG2 cells^[3].

Ref:

[1] Rossi, G., Manfrin, A., & Lutolf, M.P. (2018). Progress and potential in organoid research.

Nature Reviews Genetics, 19, 671–687. <https://doi.org/10.1038/s41576-018-0051-9>

[2] Hu, Y., Hu, X., Luo, J., et al. (2023). Liver organoid culture methods. *Cell Bioscience*, 13, 197.

<https://doi.org/10.1186/s13578-023-01136-x>

[3] Basharat, A., Rollison, H.E., Williams, D.P., & Ivanov, D.P. (2020). HepG2 (C3A) spheroids show higher sensitivity compared to HepaRG spheroids for drug-induced liver injury (DILI).

Toxicology and Applied Pharmacology, 408, 115279.

<https://doi.org/10.1016/j.taap.2020.115279>

6. Highlight the significance of the DILIRank classification system and how it ensures the biological relevance of the chosen compounds.

■ **Response:**

Thank you for your constructive feedback.

DILIRank is the largest reference drug list ranked by the risk for developing DILI in humans. The DILIRank dataset includes DILI risk classification, which is determined based on FDA-approved drug label information and literature-reported causality assessments. It consists of four categories ^[1]: (1) vMost-DILI concern: drugs withdrawn due to DILI, or drugs with black box warnings or labels containing warnings, precautions, or descriptions of severe or moderate liver injury, as validated through causality assessment. (2) vLess-DILI concern: drugs assessed as low risk based on drug labels, confirmed through causality validation. (3) Ambiguous DILI concern: drugs evaluated as high or low risk based on drug labels but lacking sufficient evidence of causality. (4) vNo-DILI concern: drugs with no literature reports confirming their role in causing DILI. Drug labeling serves as a critical basis for DILI risk classification and is derived from a systematic evaluation of preclinical toxicology data, clinical trials, post-marketing surveillance, and literature data. It provides essential drug safety information, including DILI risk, and is regarded as the “most reliable data source” ^[2].

In this study, we selected the three DILI risk annotations-vMost, vLess, and vNo-DILI-Concern-as outcome variable labels for the DILI prediction model. We filtered out “Ambiguous DILI concern” category to reduce the uncertainty caused by the lack of clear causal evidence, which enhances the robustness of deep learning modeling. Based on existing databases and literature, we included 30 drugs belongs to the Most, Less, and No-DILI categories. These 30 drugs encompass the four well-recognized mechanisms of DILI, including mitochondrial injury, reactive metabolites, biliary transport inhibition, and immune responses ^[3]. In summary, by selecting drugs categorized as Most, Less, and No-DILI concern, we ensured that these compounds are directly relevant to real-world DILI risk. Their mechanisms are widely recognized in the academic community and have direct clinical significance. Therefore, the analysis of these compounds guarantees the biological and clinical relevance of the study.

We have added corresponding content to the revision and marked them in yellow. Thank you again for your valuable input.

Ref:

- [1] Chen, M., Liao, T.-J., Li, D., et al. (2024). DILIRank dataset for QSAR modeling of drug-induced liver injury. In H. Hong (Ed.), *QSAR in safety evaluation and risk assessment* (pp. 235–243). Academic Press. <https://doi.org/10.1016/B978-0-443-15339-6.00014-X>
- [2] Chen, M., Suzuki, A., Thakkar, S., et al. (2016). DILIRank: the largest reference drug list ranked by the risk for developing drug-induced liver injury in humans. *Drug discovery today*, 21(4), 648 – 653. <https://doi.org/10.1016/j.drudis.2016.02.015>
- [3] Baudy, A. R., Otieno, M. A., Hewitt, P., et al. (2020). Liver microphysiological systems development guidelines for safety risk assessment in the pharmaceutical industry. *Lab on a chip*, 20(2), 215–225. <https://doi.org/10.1039/c9lc00768g>

■ **Revised manuscript (Page 16-17, Line 366-382):**

DILIRank is the largest reference drug list ranked by the risk for developing DILI in humans. The DILIRank dataset includes DILI risk classification, which is determined based on FDA-approved drug label information and literature-reported causality assessments. It consists of four categories:

(1) vMost-DILI concern: drugs withdrawn due to DILI, or drugs with black box warnings or labels containing warnings, precautions, or descriptions of severe or moderate liver injury, as validated through causality assessment; (2) vLess-DILI concern: drugs assessed as low risk based on drug labels, confirmed through causality validation; (3) Ambiguous DILI concern: drugs evaluated as high or low risk based on drug labels but lacking sufficient evidence of causality; (4) vNo-DILI concern: drugs with no literature reports confirming their role in causing DILI.

Drug labeling serves as a critical basis for DILI risk classification and is derived from a systematic evaluation of preclinical toxicology data, clinical trials, post-marketing surveillance, and literature data. It provides essential drug safety information, including DILI risk, and is regarded as the “most reliable data source”²¹. Herein, we categorized the data into three groups based on DILIRank classification, i.e. vMost-, vLess, and vNo-DILI concern. Notably, from the perspective of practical application, we excluded “Ambiguous DILI concern” category to reduce the uncertainty caused by the lack of clear causal evidence.

7. Discuss limitations or challenges, if any, in the methods used, briefly, like challenges faced in the experimental or computational setup.

■ **Response:**

Thank you for your constructive feedback.

From the perspective of AI design, in this study, we used the BEiT-V2 model based on the Transformer architecture to extract features by dividing images into smaller patches. While this approach performs well in many tasks, it has limitations, particularly when spatial information is processed as a whole feature, which may lead to the loss of local details, especially in biological applications that require precise capture of spatial structures;

From the perspective of biological design, in this study, due to considerations of practical applicability, convenience, and predictive performance, we selected continuously captured brightfield images of organoids at 10*Cmax as the data modality to construct a DILI prediction model, which has demonstrated strong predictive performance. However, the use of a single concentration at 10*Cmax inevitably overlooks some biological effects. In the future, we plan to incorporate a broader range of biochemical markers to capture more comprehensive DILI mechanisms and introduce a concentration gradient to better assess dose-dependent effects. This enhanced model will provide deeper mechanistic insights into liver injury, facilitating a clearer understanding of prediction results and ultimately contributing to safer drug development and more informed clinical decision-making.

RESULTS

The results are detailed, with clear comparisons between the two experimental platforms (HLOs and HepG2 spheroids).

1. Highlight the 90.16% accuracy for vNo-DILI cases earlier to draw attention to the model's strong points.

■ **Response:**

Thank you for your constructive feedback. We have moved the description of the 90.16% accuracy for vNo-DILI cases to the end of the first result section, which outlines the workflow for developing the DILI-level prediction model. This revision aims to emphasize the model's strong performance in identifying compounds with no DILI risk earlier in the results, showcasing its ability to accurately distinguish non-hepatotoxic compounds. We truly appreciate your insightful suggestions, and we believe these changes enhance the manuscript's clarity and structure.

■ **Revised manuscript (Page 3, Line 101-105):**

... The corresponding sequence of bright field images was input into the AI model to obtain the predicted value of DILI-Level. The model of this work exhibited an overall accuracy of 82.34%, with a particularly impressive performance in the vNo-DILI-concern category, where it achieved an accuracy of 90.16% (Table. 1 & Fig. 4a, c). This highlights the model's exceptional ability to identify compounds with no DILI risk, ensuring a high degree of reliability in distinguishing non-hepatotoxic compounds. More detailed comparisons will be discussed in the subsequent sections.

2. Present the metrics (accuracy, recall, specificity, F1 score) for all categories in a table for better visualization and comparison.

■ **Response:**

Thank you for your valuable feedback. We have organized the metrics (accuracy, recall, specificity, precision, and F1 score) for all categories into a table to provide a more direct comparison of the performance between the Human Liver Organoids and HepG2 Spheroids models. The updated table is shown below:

Table 1. Predictive Performance Metrics Comparison between two *in vivo* 3D Platform

Metrics		Human Liver Organoids	HepG2 Spheroids
Recall	vNo-DILI-Concern	0.9016	0.5800
	vLess-DILI-Concern	0.7500	0.6058
	vMost-DILI-Concern	0.7808	0.8580
Specificity	vNo-DILI-Concern	0.8557	0.9416
	vLess-DILI-Concern	0.9552	0.9243
	vMost-DILI-Concern	0.9059	0.6429
Precision	vNo-DILI-Concern	0.7971	0.5370
	vLess-DILI-Concern	0.7500	0.6923

	vMost-DILI-Concern	0.8769	0.6429
F1 Score	vNo-DILI-Concern	0.8462	0.5577
	vLess-DILI-Concern	0.7500	0.6462
	vMost-DILI-Concern	0.8261	0.8463
Accuracy		0.8234	0.7741

DISCUSSION

The discussion incorporates quantitative metrics like accuracy, recall, specificity, and F1 scores, providing a robust evaluation of the model's performance.

1. Start with a summary of the key findings (e.g., the superior performance of HLOs, the predictive accuracy, and clinical relevance). Follow with a comparison to previous studies, highlighting advancements and differences. Expand clinical relevance.

■ **Response:**

We greatly appreciate your insightful feedback. In response, we have revised the discussion to begin with a succinct summary of our key findings. To be specific, we emphasize the superior performance of HLOs in predicting DILI, achieving an accuracy of 82.34%, and highlight the clinical relevance of our model. Further, we expanded the comparison with previous study utilizing approach based on fluorescence, and underscored the advantages of our DILI prediction model in terms of predictive accuracy, non-invasive nature, and the incorporation of clinical data from the FDA DILIRank database. The revised manuscript as followed:

■ **Revised manuscript (Page 12-13, Line 241-279):**

(Summary of Key Findings):

In this study, we successfully developed a DILI prediction model based on organoids, which we named 'DILITracer' to highlight its ability to 'trace' the DILI level (Most-, Less-, or No-DILI). Our model achieved an average accuracy of 82.34%, demonstrating improved predictive performance for DILI prediction compared to HepG2 spheroids and animal models. Almost all of indicators (recall, specificity, precision, and F1 score) of each classification label exhibited a better value in prediction model using HLOs imaging compared to HepG2 spheroids. Our organoid experimental platform has been demonstrated to effectively mimic cell-cell interactions and exhibit higher levels of functional cytochrome P450 enzymes, suggesting that organoids serve as a more physiologically relevant in vitro 3D liver model compared to HepG2 spheroids. Furthermore, the generation of a comprehensive series of image data capturing detailed morphological features of organoids could provide a convenient and effective approach to reflect more realistic toxicological response, thereby facilitating the establishment of robust AI models. Importantly, our model has the potential to identify certain "clinically specific toxic drugs" that induce liver toxicity clinically, despite having passed standard preclinical toxicology evaluations using animal models prior to first-in-human administration. Specifically, our model successfully identified simvastatin and stavudine as "non-No-DILI" cases, which had been poorly predicted by hepatic spheroids in previous study²⁴. This may be partly attributed to the clinical relevance of the labels used in our model, where we employed clinical data-based drug classifications from the

FDA DILIRank database for model training. This approach ensures that our model is closely aligned with clinical reality, highlighting its significant practical value in clinical drug development and safety assessment. Overall, we believe it qualifies as a suitable tool for predicting DILI during preclinical drug development.

(Comparison to Previous Studies):

A previous study has developed a DILI prediction strategy based on features of fluorescence images of PHHs analyzed by a random forest algorithm²⁵. However, the fluorescent dye or probe, as an invasive way, was limited to detection at endpoints of toxicity. Therefore, we collected non-destructive brightfield images across different time series (once a day for a total of 4 days) to realize the ‘dynamic monitoring’ when clarifying the DILI toxicity. Currently, describing the structure of organoids with high phenotypic complexity using traditional morphological features such as radius length, area, and perimeter is challenging. Deep learning, however, offers a viable solution by effectively capturing the intricate patterns and features of organoids²⁶⁻²⁸. To date, for the construction of the DILI prediction model, neither 2D nor 3D imaging technologies have been used in combination with CV techniques based on deep learning. In this study, we referred to the technical principles of video processing to fully excavate the spatial and temporal features of brightfield images. These two features are extremely important for us to generate a DILI image-only model with a favorable predictive performance, as evidenced by ablation experiments: the accuracy of our establishing model (82.34%) is extremely higher than the model without spatial feature (76.10%) and model without temporal feature (70.25%). Also, we categorized input labels into three groups of DILIRanks (vMost-, vLess-, and vNo-DILI concern) based on confirmed causal evidence in clinical linking a drug to liver injury, providing a more nuanced assessment of hepatotoxicity. To the best of our knowledge, this is the first model to output ternary classification of hepatotoxicity rather than simply indicating whether or not hepatotoxicity is present.

2. Elaborate on how this model could influence preclinical testing workflows and reduce the incidence of clinical trial failures due to DILI. You can discuss potential integration with existing drug safety pipelines.

■ **Response:**

Thank you for your valuable suggestion. In response, we have expanded our discussion on the potential impact of our model on preclinical testing workflows, with the additions highlighted in yellow in the revised manuscript. The revised section is as follows:

■ **Revised manuscript (Page 13-14, Line 280-301):**

The integration of our HLO-based DILI prediction model into preclinical testing workflows has the potential to revolutionize drug safety assessment. By providing an early-stage, *in vitro* platform for hepatotoxicity evaluation, our model might significantly reduce reliance on animal models, which often struggle to predict DILI, particularly idiosyncratic DILI³⁰. Compared to previous DILI prediction models that rely on chemical structure³¹⁻³³ or gene expression^{34, 35} as data modalities, our approach offers a more convenient data acquisition process. Moreover, the early identification of clinically relevant toxic drugs during preclinical testing enables the detection of compounds with a high risk of liver toxicity before they advance to human trials, ultimately reducing costly late-stage failures. In this

study, our model’s high accuracy in identifying vNo-DILI cases (90.16%) ensures that safe drugs are prioritized for clinical trials, minimizing DILI risk and improving the likelihood of success in clinical trials and new drug projects. This approach may help lower drug development costs, provide further insights into liver toxicity risks, and offer a more reliable reference for clinical decision-making. Interestingly, the attention mechanism employed in this study revealed that our model is capable of identifying critical time point for distinguishing drug effects on organoids. This might provide valuable insights into the time window of clinical toxicity efficacy, serving as an important reference for optimizing clinical monitoring and intervention strategies—an area that warrants further investigation in future studies. By integrating dynamic brightfield imaging, machine learning, and clinical data from FDA DILIRank database, our model offers an opportunity to enhance the predictive reliability of early-stage toxicity screening. Additionally, its non-invasive nature and real-time monitoring capabilities can be seamlessly incorporated into existing drug safety pipelines, facilitating more efficient drug development and the early elimination of hepatotoxic compounds. Overall, we hope to accelerate the transition of DILI prediction model using organoids ‘from the bench to the bedside’.

3. Presentation of Data: Include more visual aids, such as tables or concise charts, to summarize key results (e.g., comparison of HLO and HepG2 spheroid performance, ablation study results, or enzyme expression levels).

■ **Response:**

Thank you for your valuable feedback. We have summarized the metrics (accuracy, recall, specificity, precision, and F1 score) in tables for a direct comparison of the Human Liver Organoids and HepG2 Spheroids models. The ablation experiment results are also presented in a table for clarity and detail. To maintain the logical flow and clarity of the article, we have placed these tables in the Results section rather than the Discussion section. We have also included appropriate interpretation in the Discussion, ensuring overall coherence and strengthening the argumentation of the study.

■ **Revised manuscript**

Result section:

Table 1. Predictive Performance Metrics Comparison between two *in vivo* 3D Platform

Metrics		Human Liver Organoids	HepG2 Spheroids
Recall	vNo-DILI-Concern	0.9016	0.5800
	vLess-DILI-Concern	0.7500	0.6058
	vMost-DILI-Concern	0.7808	0.8580
Specificity	vNo-DILI-Concern	0.8557	0.9416
	vLess-DILI-Concern	0.9552	0.9243
	vMost-DILI-Concern	0.9059	0.6429
Precision	vNo-DILI-Concern	0.7971	0.5370
	vLess-DILI-Concern	0.7500	0.6923

	vMost-DILI-Concern	0.8769	0.6429
F1 Score	vNo-DILI-Concern	0.8462	0.5577
	vLess-DILI-Concern	0.7500	0.6462
	vMost-DILI-Concern	0.8261	0.8463
Accuracy		0.8234	0.7741

Table 2. Predictive Performance Metrics in ablation experiments

Metrics		HLO without spatial	HLO using D0~D1 images	HLO
Recall	vNo-DILI-Concern	0.9016	0.7049	0.9016
	vLess-DILI-Concern	0.6200	0.7500	0.7500
	vMost-DILI-Concern	0.6986	0.6849	0.7808
Specificity	vNo-DILI-Concern	0.7835	0.7835	0.8557
	vLess-DILI-Concern	0.9626	0.9043	0.9552
	vMost-DILI-Concern	0.8706	0.7882	0.9059
Precision	vNo-DILI-Concern	0.7237	0.6719	0.7971
	vLess-DILI-Concern	0.7500	0.6923	0.7500
	vMost-DILI-Concern	0.8095	0.7353	0.8769
F1 Score	vNo-DILI-Concern	0.8029	0.6880	0.8462
	vLess-DILI-Concern	0.6818	0.7199	0.7500
	vMost-DILI-Concern	0.7500	0.7092	0.8261
Accuracy		0.7610	0.7025	0.8234

Discussion section (Page 12, Line 244-246)

Our model achieved an average accuracy of 82.34%, demonstrating improved predictive performance for DILI prediction compared to HepG2 spheroids and animal models. Almost all of indicators (recall, specificity, precision, and F1 score) of each classification label exhibited a better value in prediction model using HLOs imaging compared to HepG2 spheroids.

Discussion section (Page 13, Line 272-275)

... These two features are extremely important for us to generate a DILI image-only model with a favorable predictive performance, as evidenced by ablation experiments: the accuracy of our establishing model (82.34%) is extremely higher than the model without spatial feature (76.10%) and model without temporal feature (70.25%). ...

CONCLUSION

1. Summarize the overall contribution of the study in a concise paragraph, emphasizing the novelty of the approach, its practical relevance, and the potential for future refinement.

■ **Response:**

Thank you for your valuable suggestion. In response, we have revised the manuscript and marked them in yellow in the revised manuscript. The revised section is as follows:

■ **Revised manuscript (Page 14, Line 309-315):**

Conclusion

In this study, we successfully developed DILITracer, a DILI prediction model that analyzes spatiotemporal features from continuously captured brightfield images of liver organoids under various DILI conditions. The model correlates organoid morphology with DILI severity, providing risk assessments for compounds categorized as most-, less-, or no-DILI. DILITracer demonstrates impressive accuracy in predicting DILI levels and incorporates clinical data as outcome variables, ensuring strong clinical relevance. This AI-driven system offers a rapid and reliable tool for predicting hepatotoxicity in early-stage drug development and provides valuable insights for clinical drug screening.

Reviewer #2 (Remarks to the Author):

The authors present a tool for predicting drug-induced liver injury utilizing AI image analysis performed on liver organoid brightfield images. The authors describe this tool (DILTracer) as promising for predicting liver injury of drugs in development in their pre-clinical stage, utilizing less invasive methods such as BF imaging and AI computer vision. After establishing their model, the results of this analysis provide a ternary classification of hepatotoxicity based on the FDA DILI labels.

The authors present a clear objective of their work and their results have good interpretability in regards to its potential clinical application.

This work also adds another application for utilizing organoids as in-vitro models for drug development and clinical correlation proving the various advantages of organoids due to their characteristics of maintaining fidelity to their source and adaptability for high-throughput testing. The application of AI analysis on brightfield images of various tissue organoids is not novel and has been reported in other literature to quantify the growth of organoids. The authors do present this point and argue that drug-induced liver injury has not been of major focus. Therefore, this work does present a potentially promising benefit as an add-on to current methods of pre-clinical drug testing.

In this revised manuscript, the authors addressed my comments from the first review. Figure 2 includes larger photos showcasing the HLO's under brightfield imaging under different treatments.

The re-organization of the text makes it smoother to read and follow the sequence of the experiments and results. The discussion section highlights some important limitations of the study but also in my opinion promising applications of this tool such as the ability to identify critical time points of the drug effects on the organoids given the non-invasive nature of BF imaging.

I have no further comments or reviews.

Response to Reviewer #2:

We sincerely thank the reviewer for their thoughtful and constructive comments in the previous round of review. We are grateful that the revisions in the current manuscript were found to be satisfactory. We appreciate your recognition of the clarity of our objectives, the interpretability of our results, and the potential clinical relevance of our tool. Thank you again for your support and encouraging feedback.

Reviewer #3 (Remarks to the Author):

Thank you for your hard work on this manuscript. I have reviewed the latest version, and all suggested changes have been successfully incorporated. The manuscript is now in great shape and ready for publication.

Response to Reviewer #3:

We thank the reviewer for their positive evaluation of our revised manuscript. We are pleased to know that all suggested changes have been successfully addressed and that the manuscript is now considered ready for publication. We truly appreciate your support.